# AdaViewPlanner: Adapting Video Diffusion Models for Viewpoint Planning in 4D Scenes

**Yu Li**[1]* **Menghan Xia**[2]† **Gongye Liu**[4] **Jianhong Bai**[5] **Xintao Wang**[3]
**Conglang Zhang**[6] **Yuxuan Lin**[1] **Ruihang Chu**[1] **Pengfei Wan**[3] **Yujiu Yang**[1]†

[1]Tsinghua University [2]HUST [3]Kling Team, Kuaishou Technology [4]HKUST
[5]Zhejiang University [6]Wuhan University

https://yuli0103.github.io/AdaViewPlanner/

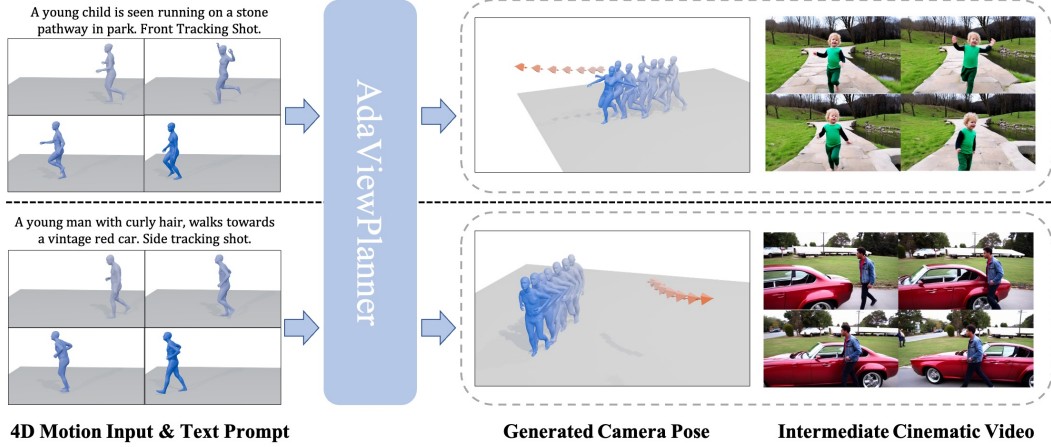

4D Motion Input & Text Prompt     Generated Camera Pose     Intermediate Cinematic Video

Figure 1: Showcasing AdaViewPlanner: Given 4D contents and text prompts that depicts scene context and desired camera movements, we adapt pre-trained video diffusion models to generate coordinate-aligned camera pose sequence as well as an corresponding video visualization.

## Abstract

Recent Text-to-Video (T2V) models have demonstrated powerful capability in visual simulation of real-world geometry and physical laws, indicating its potential as implicit world models. Inspired by this, we explore the feasibility of leveraging the video generation prior for viewpoint planning from given 4D scenes, since videos internally accompany dynamic scenes with natural viewpoints. To this end, we propose a two-stage paradigm to adapt pre-trained T2V models for viewpoint prediction, in a compatible manner. First, we inject the 4D scene representation into the pre-trained T2V model via an adaptive learning branch, where the 4D scene is viewpoint-agnostic and the conditional generated video embeds the viewpoints visually. Then, we formulate viewpoint extraction as a hybrid-condition guided camera extrinsic denoising process. Specifically, a camera extrinsic diffusion branch is further introduced onto the pre-trained T2V model, by taking the generated video and 4D scene as input. Experimental results show the superiority of our proposed method over existing competitors, and ablation studies validate the effectiveness of our key technical designs. To some extent, this work proves the potential of video generation models toward 4D interaction in real world.

---

*This work was conducted during the author's internship at Kling Team, Kuaishou Technology.
†Corresponding authors.

# 1 INTRODUCTION

Digital animation has crucial applications in fields like gaming, education, and film. Dynamic 3D content, or 4D scenes, can be created in various ways—such as virtual modeling, scanning or reconstruction from real-world scene. Ultimately, these scenes require carefully designed virtual camera viewpoints to be rendered as engaging videos. However, manually arranging the camera shots and movements for specific targets is both tedious and requires specialized expertise. Therefore, there is a high demand for automatic cinematography generation techniques that can operate based on given 4D content and instructions.

Existing research (Wang et al., 2024c; Courant et al., 2024) typically relies on specialized models trained on limited datasets. While these methods have achieved impressive results in specific scenarios, they often struggle to generalize to open-world scenes and lack support for preference controls, such as text instructions. Inspired by the powerful capabilities of recent text-to-video (T2V) models, we explore the feasibility of repurposing such models as virtual cinematographers to design professional camera trajectories for 4D scenes. Our key insight is that pre-trained T2V models can generate vivid dynamic content with professional camera movements based on text prompts, which indicates their internal knowledge of how to match camera movements to dynamic scenes. Importantly, their vast training data provides strong generalization to various scenes, and their text-following ability can be inherited naturally.

Building on this insight, we propose to leverage the cinematographic expertise embedded within pre-trained video generation models. To inherit this generative prior smoothly, we designed a two-stage paradigm, where each stage integrates the original video generation path with a conditional control or parameter prediction. Firstly, we use an adaptive learning branch to inject a 4D scene representation into a pre-trained text-to-video (T2V) model. While the 4D scene itself is viewpoint-agnostic, the video generated from it visually embeds the target camera viewpoints. However, the ambiguous projecting relationship between the rendered video and the 4D content can lead to training collapse. To prevent this, we introduce a guided learning scheme that randomly provides ground-truth camera poses to the model as hints. This helps the model understand the 4D input and synthesize video content with plausible camera movements. Second, we formulate viewpoint extraction as a hybrid-condition guided camera parameter denoising process. We introduce a dedicated camera diffusion branch to the motion-conditioned T2V model, which takes both the generated video and the 4D scene as input. Through this two-stage method, we can obtain a sequence of camera poses that are aligned with the input 4D scene's coordinate system, along with a corresponding video that visualizes the 4D scene from the predicted camera viewpoints.

Experimental results illustrate that our proposed method outperforms existing competitors by a large margin, thanks to the powerful generative capability of pre-trained foundation models. In addition, extensive ablation studies are conducted to validate the effectiveness of our key technical designs, which explains how the advantages of our method were achieved. Our contributions can be summarized as follows.

- We are the first to explore adapting pre-trained T2V models for viewpoint planning in 4D scenes, which offers advantages in open-world generalization and prompt-following.
- We propose a novel two-stage method that leverages the video generation prior to arrange camera poses based on conditional 4D content in a compatible manner.
- Our work offers a promising proof of concept for the potential of using video generation models as "world models" for 4D interaction.

# 2 RELATED WORKS

**Camera Planning.** Automated camera planning, or computational cinematography, seeks to generate optimal camera trajectories for virtual scenes to enhance storytelling and user experience (Zhang et al., 2025; Jiang et al., 2024; Rao et al., 2023). Early data-driven methods (Jiang et al., 2020; 2021; Hou et al., 2024) relied on reference-based frameworks, while recent approaches leverage deep generative models to synthesize novel trajectories from diverse inputs. A key line of work focuses on character-driven camera motion, e.g., imitation learning for drone filming (Huang et al., 2019; Wang et al., 2023b) and GAN-based planning in interactive environments (Yu et al.,

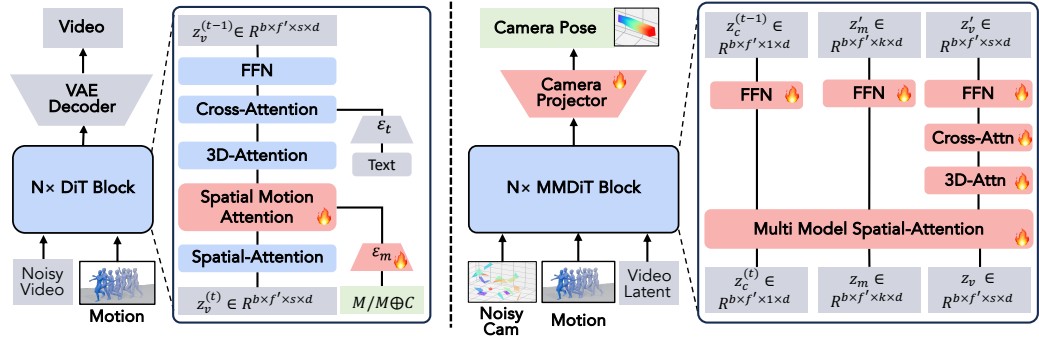

**Figure 2:** (a) Stage I model for motion-conditioned cinematic video generation: a pose encoder processes human motion data ($M$) from 4D scenes and integrates it with video tokens via spatial motion attention to produce videos with cinematic camera movements. Camera parameters used for guidance are denoted as $C$. (b) Stage II model: three branches for video, camera, and human motion are combined in an MMDiT framework to extract camera pose.

2023). Multi-modal extensions include DanceCamera3D (Wang et al., 2024c) and Dancecamanimator (Wang et al., 2024d), which conditions camera motion on dance and music. Text-conditioned generation has also emerged, with models such as Director (Courant et al., 2024) and Director3D (Li et al., 2024) enabling intuitive control and 3D scene synthesis. Related generative techniques further extend to domains like robot navigation (Bar et al., 2025), demonstrating the broad applicability.

**Human Motion Control for VDMs.** Recent work in controllable human video generation has progressed from 2D pose guidance (Hu, 2024; Xu et al., 2024; Peng et al., 2024) to leveraging 3D motion conditions (Zhu et al., 2024; Ding et al., 2025; Shao et al., 2024), which resolve depth ambiguity and self-occlusion by providing explicit geometric information. A key challenge is integrating such 3D structure into generative models like diffusion. Some methods incorporate raw 3D motion directly, e.g., MTVCrafter (Ding et al., 2025) and ISA-4D (Shao et al., 2025), while others adapt pre-trained models through sparse keypose control (Guo et al., 2025). Beyond motion alone, Uni3C (Cao et al., 2025) and RealisMotion (Liang et al., 2025) embed characters into coherent 3D scene or world coordinate spaces, enabling unified control of motion and camera. These works underscore the growing importance of explicit 3D representations for human video synthesis. However, prior methods either rely on fixed-viewpoint 2D human pose sequences as input, or require 4D human poses together with camera parameters. Our approach, by comparison, conditions only on normalized 4D human poses, while the video model itself plans the viewpoints and camera motions.

## 3 METHOD

Training over vast amounts of film footage, video generation models can synthesize various dynamic scenes with rich cinematic skills. Based on this observation, we aim to leverage this capability by repurposing these models as virtual cinematographers to design professional camera trajectories for given 4D scenes. For simplicity, we explore and validate this concept by only considering moving human in 4D scenes, which serves as the major context of interest in applications.

**Problem Formulation.** Given a sequence of human motion represented as SMPL-X (Pavlakos et al., 2019) 3D joint positions $M \in \mathbb{R}^{f \times k \times 3}$, , where $f$ denotes the number of frames and $k = 22$ represents the selected body joints, our objective is to generate a corresponding camera trajectory $C \in \mathbb{R}^{f \times 9}$, so that the scene content could be visualized as a video with professional and plausible camera movements. The camera parameters for each frame consist of a 3D translation vector $(t_x, t_y, t_z)$ and a 6D rotation representation using the first two rows of the rotation matrix $(R_{11}, R_{12}, R_{13}, R_{21}, R_{22}, R_{23})$, which can be orthogonalized to obtain a valid rotation matrix.

**Overview.** To smoothly leverage the prior knowledge embedded in pre-trained text-to-video (T2V) models (Chen et al., 2024; Wan et al., 2025), we propose a two-stage approach. Firstly,

we equip a pre-trained video generation model with an adaptive learning branch to synthesize cinematographically-informed video content based on human motion. This process effectively determines the target camera trajectories through video rendering. Secondly, we formulate camera parameter extraction as a hybrid-condition guided camera extrinsic denoising process. To do this, we introduce a new camera extrinsic diffusion branch to the motion-conditioned T2V model, which takes the generated video and skeleton sequence as input. Since our approach inherits the capabilities of the pre-trained T2V model, text prompts can be used to control both the scene's context and the camera's movement style. In essence, this feature enriches the conditional representation of 4D scene, which is otherwise based solely on human skeletons.

## 3.1 STAGE I: MOTION-CONDITIONED CINEMATIC VIDEO GENERATION

In the first stage, we train a video generation model to autonomously design the camera trajectory for a given 4D human motion sequence. Unlike conventional human animation approaches that operate from fixed viewpoints using 2D human pose sequences, or recent methods (Shao et al., 2025; Li et al., 2025a) that require explicit camera parameters as additional input, our approach takes only a normalized 4D human motion sequence as input, and leverages learned cinematographic priors to determine optimal camera viewpoints and movements.

**Spatial Motion Attention.** Inspired by 3DTrajMaster (Fu et al., 2024), we inject the motion condition through a spatial motion attention mechanism integrated within the DiT (Peebles & Xie, 2023) framework. Specifically, a pose encoder first maps the input motion sequence $M \in \mathbb{R}^{f \times k \times 3}$ into a latent embedding $z_m \in \mathbb{R}^{f' \times k \times d}$, where temporal downsampling modules is adopted to align the temporal resolution $f'$ with the VAE-encoded video latents. Subsequently, we concatenate the video tokens $z_v^{(t)}$ with the human motion tokens $z_m$ along the spatial dimension before feeding them into the self-attention block, leveraging the natural frame-wise correspondence.

$$T = [z_v^{(t)}; z_m] \in \mathbb{R}^{f' \times (h \cdot w + k) \times d} \tag{1}$$

Standard self-attention is then applied to the combined sequence:

$$q = W_Q \cdot T, \quad k = W_K \cdot T, \quad v = W_V \cdot T$$
$$z_v^{(t)} = z_v^{(t)} + \text{Truncate}\left(\text{Attn}(q, k, v)\right) \tag{2}$$

where Truncate discards the human motion token outputs and retains only the updated video tokens.

**Guided Learning Scheme.** We found that generating cinematically appealing videos from motion alone is challenging. This direct, unguided learning requires the model to simultaneously understand 3D human dynamics and cinematographic principles, then render motion-consistent 2D videos with implicit camera trajectories. To address this complexity, we introduce a curriculum learning strategy. With probability $p$, we provide the model with explicit camera information $z_c$, obtained by encoding the camera pose through an independent pose encoder structurally analogous to the human pose encoder, creating a combined token sequence for joint camera-motion control.

$$T = \begin{cases} \left[z_v^{(t)}; z_m; z_c\right] \in \mathbb{R}^{f' \times (hw + k + 1) \times d}, & \text{with prob. } p, \\ \left[z_v^{(t)}; z_m\right] \in \mathbb{R}^{f' \times (hw + k) \times d}, & \text{with prob. } 1 - p. \end{cases} \tag{3}$$

This guidance helps the model learn to render videos that conform to given human motion under given camera views before tackling autonomous camera design, effectively reducing the training complexity. Similar to MTVCrafter (Ding et al., 2025), we use 3D spatial RoPE (Su et al., 2024) to encode the human motion tokens and employ pose-specific RoPE encodings to differentiate between motion and camera tokens.

During this stage, we freeze the base video model and exclusively train the newly introduced human motion encoder and the spatial motion attention layers.

## 3.2 STAGE II: CAMERA POSE EXTRACTION

The model trained in the first stage generates videos with implicit, model-designed cinematography for given 4D human motion sequences. In the second stage, camera poses are explicitly extracted and aligned with the reference 4D human motion coordinate system.

While existing camera estimation methods (Li et al., 2025b; Zhang et al., 2024; Wang et al., 2024a) can extract camera poses from video, they typically face two critical limitations in our context: (1) they require complex post-processing to align the estimiated camera pose with the human motion coordinate frame, and (2) AI-generated videos often contain geometric and texture inconsistencies that compromise feature-matching-based estimation, leading to trajectory jitter, fragmented camera paths, and failures in scene reconstruction, as shown in Figure 11.

To address these challenges, we train a direct estimation model using paired human motion and video sequences to predict absolute camera poses within the reference human coordinate system. We adopt the MMDiT framework (Esser et al., 2024) with three specialized branches to accommodate the distinct characteristics of our multi-modal input—the video containing the cinematic information and the human motion providing the reference coordinate frame, as illustrated in Figure 2(b). The video branch is initialized from the pre-trained video model, while the camera and human motion branches are randomly initialized with simplified architectures consisting of spatial attention and FFN layers. Similar to Stage I, we employ pose encoders to encode the camera and motion inputs.

We adopt a flow-matching (Lipman et al., 2022) objective, training the model to predict the vector field that transports noisy camera parameters toward the clean ones (Wang et al., 2023a). During training, video tokens $z_v$ and human motion tokens $z_m$ serve as clean conditions while camera tokens $z_c^{(t)}$ linearly combine with noise depending on the randomly sampled timestep. The multi-modal spatial attention operates on concatenated tokens along the spatial dimension:

$$q = [q_v; q_m; q_c^{(t)}], \quad k = [k_v; k_m; k_c^{(t)}], \quad v = [v_v; v_m; v_c^{(t)}] \tag{4}$$

For training, we employ synthetic data rendered from Unreal Engine (UE) (Epic Games, 2022), which provides diverse human motions along with precise camera parameters. Furthermore, we employ GVHMR (Shen et al., 2024) to reconstruct the 4D human motion and unify the camera and human motion data into a common coordinate system.

## 4 EXPERIMENTAL RESULTS

### 4.1 EXPERIMENT SETTINGS

**Implementation Details.** Both the Stage I and Stage II models are initialized from a pretrained 1B Transformer T2V backbone built for internal research. For the Stage I model, we first train on 400k unfiltered videos of resolution $384 \times 672$ for 15k iterations, followed by fine-tuning on 10k curated high-quality internal videos with camera motion for another 10k iterations. Training employs the Adam optimizer on 16 NVIDIA H800 GPUs with a total batch size of 64, a learning rate of $5 \times 10^{-5}$, a timestep shift of 15, and the probability of using camera guidance is 0.5. The Stage II model is trained on 101k MultiCamVideo (Bai et al., 2025), 43k HumanVid UE (Wang et al., 2024b), and 100k internal UE videos. It proceeds in two phases: first, a model is trained to predict relative camera poses for 10k iterations; then, based on this model, we introduce a human motion branch and further train it to predict absolute camera poses for 40k iterations. Here, the timestep shift is reduced to 1, while other hyperparameters remain unchanged from Stage I. During inference, both models use 50 sampling steps. We use GVHMR (Shen et al., 2024) to reconstruct 4D human motion from video data and transform it into the canonical space. Details can be found in Appendix B.

**Baselines.** Due to the lack of existing works with an identical experimental setting, we select the following methods as our baselines for comparison. E.T. (Courant et al., 2024) generates camera motion conditioned on text and character trajectory, modeling the character as a single point whose continuous motion is represented as a point trajectory. DanceCamera3D (Wang et al., 2024c) synthesizes camera motion from audio and dance poses. The original model is incompatible with our evaluation protocol as it requires audio input and uses a different skeleton representation. Therefore, we adapt their framework to our task by replacing the audio condition with a text prompt and retraining the model on our own dataset.

**Evaluation Metrics.** Existing metrics like Fréchet Camera Distance (FCD) and Text-Camera Consistency Score (CS) from prior work (Courant et al., 2024) suffer from distributional bias and fail to capture camera-human interaction. To address these limitations, we propose a comprehensive three-part evaluation framework. **(1) Rule-based Assessment:** We adapt and significantly refine

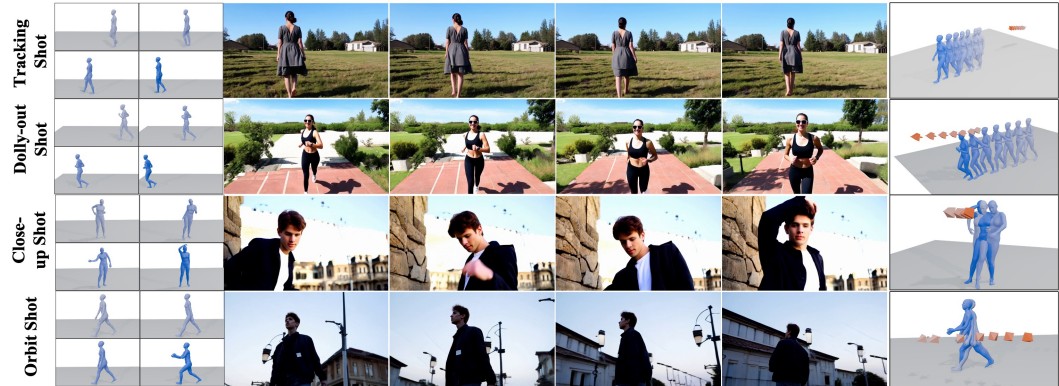

Figure 3: Visualization of results. (Left) Human motion conditions; (Middle) Stage I generated videos; (Right) Stage II generated camera trajectories. AdaViewPlanner demonstrates the ability to design diverse, instruction-consistent, and human-centered camera trajectories.

the rule-based metrics from DanceCamera3D (Wang et al., 2024c) for objective evaluation, including human visibility (Human Missing Rate), camera smoothness via jerk, and a geometrically-aware shot diversity metric capable of distinguishing different perspectives, which is an improvement over prior view-invariant methods. **(2) MLLM-based Evaluation:** Given the lack of automated metrics for evaluating high-level qualities of trajectories, we introduce an MLLM-based evaluator leveraging Gemini 2.5 Pro (Comanici et al., 2025) that analyzes orthographic trajectory visualizations, scores text-camera consistency on a 0-2 scale with detailed justification, and quantifies the diversity of cinematographic style by calculating the entropy of categorized camera attributes.

**(3) User Study:** We invited 12 researchers in computer vision–related fields to participate in our user study. Our user study consisted of two test sets: the E.T. test set with 21 samples and our own test set with 30 samples. In the questionnaire, we clearly explained the evaluation procedure and criteria to the participants. Specifically, each question presented the textual description of the camera motion along with three randomly ordered results generated by different methods. The evaluation criteria included: (1) consistency between the camera trajectory and the textual instruction, (2) professionalism of the camera motion, and (3) the coherence between the camera motion and the human actions. Each participant was asked to select the option they believed to be the best for each sample. We then aggregated all responses and computed the preference rate for each method.

Detailed evaluation protocols are provided in Appendix C.

## 4.2 COMPARISON WITH STATE-OF-THE-ART METHODS

**Qualitative Results.** Figure 3 presents the complete results of our method, showing that it can design human-centric, cinematic camera motions for diverse instructions and actions, with Stage I preview videos offering strong qualitative demonstrations. Furthermore, as illustrated in Figure 4, we analyze the effects of random seeds, scene styles, and camera instructions. The top rows show that varying the initial noise for the same human motion yields different trajectories. The middle rows show that our method adapts to various scene styles, generating camera motions that match the visual aesthetics of each environment. The bottom rows highlight the model's ability to follow camera instructions, producing multi-view and multi-motion trajectories centered on the subject.

Figure 5 compares AdaViewPlanner with other approaches. Since E.T. reduces human motion to point trajectories, it fails to capture complex, meaningful actions, resulting in simple trajectories lacking diversity and cinematic style. Our improved DanceCam*, though trained on the same dataset, struggles with convergence due to the divergent nature of the task and ultimately collapses to a single trajectory with limited diversity, as further shown in Figure 7. In contrast, AdaViewPlanner leverages video models to encode strong 4D scene priors and rich cinematic knowledge learned from data, with its two-stage pipeline effectively addressing these challenges.

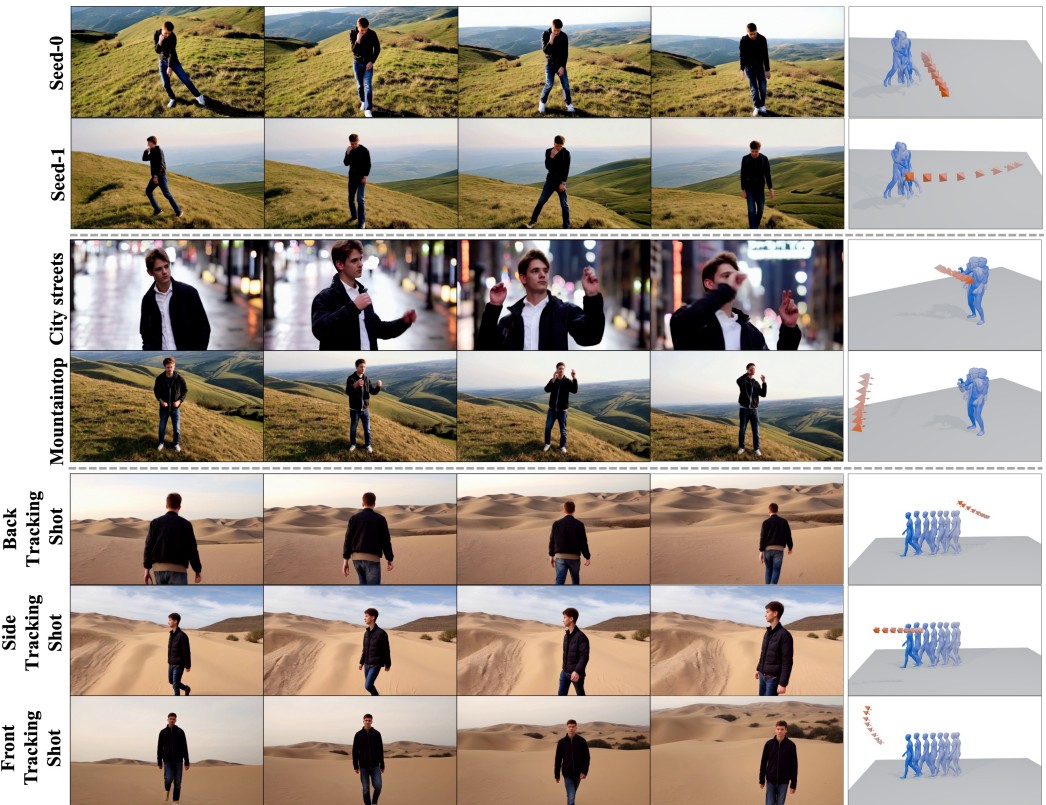

Figure 4: Camera generation results with varied random seed (**top**), scene context prompt (**middle**), and camera movement prompt (**bottom**).

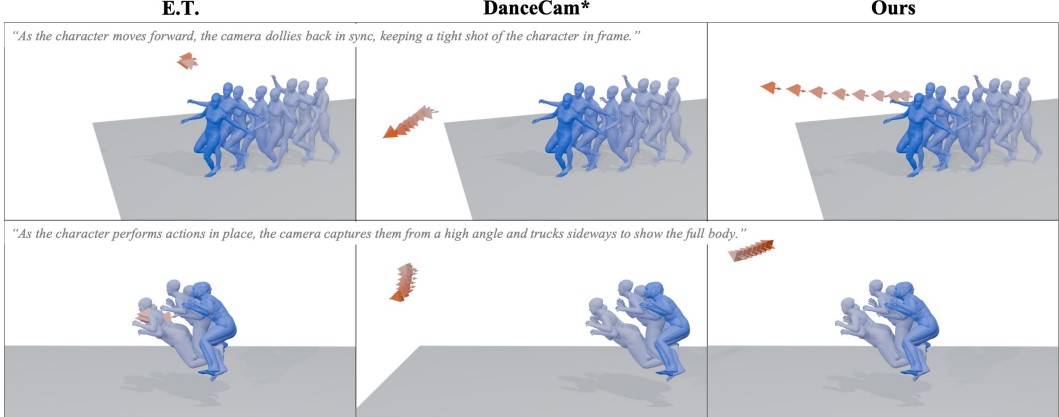

Figure 5: Compared with other methods, our model generates smoother trajectories that better follow instructions, while also exhibiting a cinematographic style centered on human actions.

**Quantitative Results.** We compare our method with baselines in Section 4.1. Evaluation is conducted on the SMPL-based test set from E.T., refined to 500 samples to reduce jitter and artifacts. We also build an internal dataset of 240 samples with higher-quality poses.

As shown in Table 1, our method outperforms all baselines on both test sets. HMR results show our generated cameras are more character-centered, while Jerk results indicate smoother trajectories. For in-place character motions, E.T. can only take a static point, leading to mostly stationary cameras and thus lower $\text{Jerk}_t$ values. However, jitter in its rotation matrices still undermines trajectory smoothness. The Dist results show that our cameras achieve greater diversity across the $360°$

Table 1: Quantitative results on the E.T. testset and our curated testset. Metrics: *HMR* (Human Missing Rate), $Jerk_t$/$Jerk_r$ (Camera Jerk of Translation/Rotation), $Dist_t$/$Dist_r$ (Shot Diversity of Translation/Rotation), *TCC* (Text-Camera Consistency), *CSD* (Cinematographic Style Diversity). DanceCam* denotes our re-implementation of DanceCamera3D using our skeleton format.

| Method | Rule-based | | | | | MLLM-based | | User Study |
|---|---|---|---|---|---|---|---|---|
| | HMR ↓ | $Jerk_t$ ↓ | $Jerk_r$ ↓ | $Dist_t$ ↑ | $Dist_r$ ↑ | TCC ↑ | CSD ↑ | Pref. (%) ↑ |
| *E.T. Testset* | | | | | | | | |
| E.T. | 0.064 | **0.001** | 0.026 | 0.538 | **0.540** | 0.850 | 0.608 | 23.81 |
| DanceCam* | 0.053 | 0.013 | 0.003 | 1.236 | 0.290 | 0.975 | 0.569 | 14.29 |
| Ours (Full) | **0.044** | 0.007 | **0.002** | **2.826** | 0.533 | **1.125** | **0.686** | **61.90** |
| *Ours Testset* | | | | | | | | |
| E.T. | 0.048 | **0.001** | 0.029 | 0.700 | 0.225 | 0.790 | 0.623 | 20.83 |
| DanceCam* | 0.024 | 0.014 | 0.002 | **1.535** | 0.189 | 0.867 | 0.593 | 15.83 |
| Ours (Full) | **0.018** | 0.003 | **0.001** | 1.415 | **0.529** | **1.385** | **0.711** | **63.33** |

Table 2: Quantitative comparison for 4D human motion control on TikTok (dance domain) and our curated general domain testsets. We report *WA-MPJPE* and *PA-MPJPE* in millimeters. Here, *Ours Subopt* denotes an early-training checkpoint included as a performance-degraded reference.

| Method | TikTok (Dance Domain) | | General Domain | |
|---|---|---|---|---|
| | WA-MPJPE ↓ | PA-MPJPE ↓ | WA-MPJPE ↓ | PA-MPJPE ↓ |
| MTVCrafter (CogVideoX-5B) | 84.89 | 22.01 | 222.50 | 38.90 |
| MTVCrafter (Wan-2.1-14B) | 73.47 | **20.22** | 224.50 | 40.21 |
| Ours Subopt | 95.06 | 30.04 | 161.22 | 40.41 |
| Ours w/o Guided Learning | 72.60 | 25.49 | 127.68 | **33.80** |
| Ours w/o 3D RoPE | 82.59 | 24.70 | 122.13 | 38.71 |
| Ours (Full) | **71.65** | 23.76 | **103.92** | 35.70 |

space around the character. Benefiting from the video model's ability to plan camera motions conditioned on text and actions, the MLLM-based evaluation confirms that our trajectories better follow textual descriptions and exhibit higher cinematographic diversity. The user study results show that our method achieved over 60% user preference across both test sets, indicating that the camera trajectories generated for 4D secenes were more favorably received.

### 4.3 MORE ANALYSIS AND ABLATION STUDIES

**Comparison of Human Motion Control.** The videos obtained in the first stage, which exhibit stronger consistency with human motion, help improve the camera results in the second stage. When there is a large discrepancy between the video and the human motion condition, the model's prediction of the camera pose is negatively affected, which is validated in Table 3.

We select MTVCrafter (Ding et al., 2025), which uses SMPL (Loper et al., 2023) poses as conditions, as our primary baseline. Although ISA4D (Shao et al., 2025) supports SMPL poses but cannot be included due to unavailable code. Since MTVCrafter was trained on dance-specific data with a fixed view, we evaluate fairly on the TikTok test set and additionally on a general human pose domain. All methods are given the same 4D motion conditions, and motion control is evaluated by MPJPE computed from GVHMR-reconstructed poses of the generated videos. While our method generates videos with camera motion and the baseline outputs fixed frontal views, Table 2 shows our method matches Wan-2.1-14B base MTVCrafter on the dance domain and significantly outperforms it on the general domain, benefiting from training on broader pose distributions. Rows 4 and 5 demonstrate the effectiveness of incorporating guidance view and 3D RoPE.

**Ablation Studies on Camera Generation.** We conduct ablation studies on Stages I and II to assess the role of our design choices. First, as shown in the first row of Table 3, the misalignment

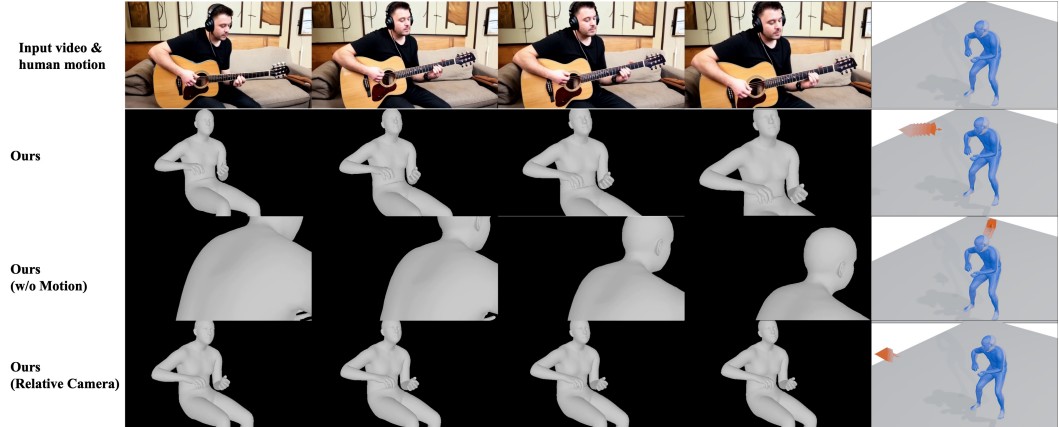

Figure 6: Columns 1–4 show the reprojection of 4D human skeletons by using estimated camera parameters, while Column 5 presents the rendered results in 3D space. *w/o Motion* exhibits viewpoint errors, whereas *Relative Cam* suffers from scale perception issues.

Table 3: Ablations on design choices for camera-trajectory generation. The *Reproject Acc* is computed by reprojecting 4D human poses and comparing against the human region mask in the original video. Variants: *Stage I Subopt* (early-training checkpoint of Stage I), *Stage II w/o Motion* (no motion conditioning in Stage II), and *Stage II Relative Cam* (Stage II predicts relative camera poses).

| Method | Rule-based | | | | | MLLM-based | | Reproject Acc | |
|---|---|---|---|---|---|---|---|---|---|
| | HMR $\downarrow$ | Jerk$_t$ $\downarrow$ | Jerk$_r$ $\downarrow$ | Dist$_t$ $\uparrow$ | Dist$_r$ $\uparrow$ | TCC $\uparrow$ | SD $\uparrow$ | MSE $\downarrow$ | IoU $\uparrow$ |
| Stage I Subopt | 0.021 | 0.003 | 0.001 | 1.222 | 0.517 | 1.051 | 0.681 | 0.181 | 0.301 |
| Stage II w/o Motion | 0.027 | 0.006 | 0.003 | 1.412 | **0.907** | 0.931 | 0.595 | 0.255 | 0.226 |
| Stage II Relative Cam | 0.039 | **0.002** | 0.001 | **1.588** | 0.518 | **1.413** | 0.698 | 0.167 | 0.325 |
| Ours (Full) | **0.018** | 0.003 | **0.001** | 1.415 | 0.529 | 1.385 | **0.711** | **0.158** | **0.338** |

Table 4: Quantitative comparison for 4D human motion control on the AMASS and GTA-Human datasets.

| Method | AMASS | | GTA-Human | |
|---|---|---|---|---|
| | WA-MPJPE $\downarrow$ | PA-MPJPE $\downarrow$ | WA-MPJPE $\downarrow$ | PA-MPJPE $\downarrow$ |
| MTVCrafter (CogVideoX-5B) | 107.57 | 49.58 | 270.75 | 55.61 |
| MTVCrafter (Wan-2.1-14B) | 103.69 | 45.78 | 240.72 | 56.46 |
| **Ours** | **72.19** | **42.13** | **145.00** | **50.23** |

of human motion between the 4D condition and generated video from Stage I would degrade the camera pose extraction accuracy of Stage II. Second, removing the motion condition in Stage II degrades performance (Figure 6): without skeletal references, the model produces smooth but misfocused trajectories, yielding erroneous viewpoints. The Reproject Acc metric underscores the need to condition on human motion to model dynamic subjects in video and guide camera focus during training. Finally, we train a variant without motion conditioning that estimates only relative camera poses, requiring post-processing to align camera and human motion coordinates. While this resolves viewpoint issues, the model lacks motion scale awareness, leading to noticeable inconsistencies.

**Discussion on Unification of Stage I and II.** We attempt to unify Stages I and II into a single model. However, this paradigm faces three challenges: (1) motion control and camera estimation conflict, since the former requires timestep sampling biased toward high-noise regions while the latter benefits from low-noise regions, leading to degraded joint performance; (2) noisy videos diminish pixel motion cues, reducing camera estimation accuracy; and (3) unified training requires real datasets annotated with both skeletal and camera parameters, but accurate camera parameter estimation from real videos remains a major bottleneck. We therefore leave this direction to future work.

**Discussion on w/o Video Model.** To assess the role of the video model in camera trajectory design for 4D scenes, we train variants that generate trajectories from text and human poses alone: one adapted from DanceCamera3D, and another based on Stage II without the video branch. In both cases, training either fails to converge or collapses to a single trajectory, as shown in Figure 7, leading to limited diversity. This highlights the divergent nature of the task, which is difficult to resolve without additional guidance. By contrast, our two-stage framework, equipped with a pretrained video model, effectively mitigates this issue and consistently produces more diverse and reliable trajectories.

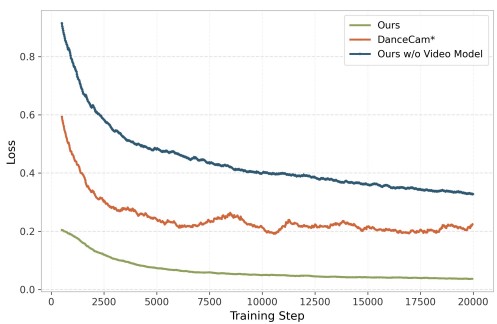

Figure 7: Comparison of training loss curves: Ours, DanceCam* and Ours w/o Video Model.

Table 5: Quantitative results of viewpoint planning on the AMASS and GTA-Human test sets.

| Method | Rule-based Metrics | | | | | MLLM-based Metrics | |
|---|---|---|---|---|---|---|---|
| | HMR $\downarrow$ | Jerk$_t$ $\downarrow$ | Jerk$_r$ $\downarrow$ | Dist$_t$ $\uparrow$ | Dist$_r$ $\uparrow$ | TCC $\uparrow$ | CSD $\uparrow$ |
| *AMASS Testset* | | | | | | | |
| E.T. | 0.033 | **0.002** | 0.015 | 0.422 | 0.150 | 0.900 | 0.564 |
| DanceCam* | 0.031 | 0.018 | 0.006 | 0.503 | 0.129 | 0.950 | 0.556 |
| **Ours** | **0.015** | 0.003 | **0.001** | **1.437** | **0.577** | **1.220** | **0.710** |
| *GTA-Human Testset* | | | | | | | |
| E.T. | 0.110 | 0.004 | 0.063 | 0.620 | 0.330 | 0.880 | 0.585 |
| DanceCam* | 0.048 | 0.021 | 0.007 | 0.810 | 0.288 | 0.980 | 0.614 |
| **Ours** | **0.039** | **0.004** | **0.001** | **1.556** | **0.675** | **1.160** | **0.729** |

## 4.4 GENERALIZATION TO REAL-WORLD DATA

The primary motivation of our work is to automate the process of professional camera choreography for 3D dynamic content within the computer graphics domain. In typical CG production pipelines, 3D human models originate from two primary sources: they are either synthetically generated (e.g., by game engines or professional designers) or captured from real-world performances using motion capture systems. To ensure our framework generalizes effectively across these distinct data types, we select two representative datasets for evaluation. Specifically, for synthetic content, we use the GTA-Human dataset (Cai et al., 2024), and for real-world captured motion, we employ the AMASS dataset (Mahmood et al., 2019). For our evaluation, we randomly sample 100 instances from the test split of each dataset. The results for 4D human motion control and viewpoint planning on real test sets are presented in Table 4 and Table 5, respectively. These comprehensive results confirm that our method generalizes robustly to diverse, real-world 4D data sources.

## 5 CONCLUSION AND LIMITATIONS

In this work, we explored adapting pre-trained T2V models for viewpoint planning in 4D scenes. Our key insight is that pre-trained video models generate realistic dynamic content accompanied by professional camera movements, revealing their internal knowledge of cinematography in dynamic environments. To leverage this prior, we design a two-stage framework: (1) an adaptive learning branch injects 4D scene representations into the pre-trained T2V model with ground-truth viewpoints guiding training; (2) a dedicated camera diffusion branch formulates viewpoint extraction as a hybrid-condition guided denoising process. Experiments show our method significantly outperforms prior approaches, and ablation studies verify the effectiveness of our core technical designs. Additionally, AdaViewPlanner has several limitations, including limited support for general 4D scenes and the inheritance of base video model shortcomings (e.g., geometric inconsistencies and challenges with complex motions), which are discussed in detail in Appendix B.

**Ethics statement.** We take research ethics very seriously and strictly adhered to the ICLR Code of Ethics throughout the user study. Before the study began, all participants were provided with a detailed informed consent form explaining the study procedures, and their explicit consent was obtained. All collected data were anonymized to protect participants' rights and privacy. Participation in the study was entirely voluntary; participants were informed that they could withdraw at any time without providing a reason, and that all their data would be deleted upon withdrawal. The study involved minimal risk, ensuring that no physical, psychological, or social harm was posed to participants. In addition, as with any generative model, our method carries the risk of potential misuse. We emphasize that the system should be applied responsibly and urge caution to avoid malicious or harmful applications.

**Reproducibility statement.** The framework and algorithms of AdaViewPlanner are presented in Sec. 3 and Appendix A. Details of training settings, training data, and hyperparameters are provided in Sec. 4.1, while inference details and data processing steps are described in Appendix B. A comprehensive introduction of the evaluation metrics is given in Appendix C, with the specifics of the MLLLM-based evaluation further discussed in Appendix D. In addition, the instruction templates used for evaluation are included in Table 6 and Table 7. Experimental setups and detailed results can be found in Sec. 4.

## ACKNOWLEDGMENTS

This work was partly supported by the National Natural Science Foundation of China (Grant Nos. 62576191 and 62502169) and the Shenzhen Science and Technology Program (ZDCY20250901103533010).

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

APPENDIX

Our Appendix consists of 7 sections. Readers can click on each section number to navigate to the corresponding section:

- Section **A** introduces the base text-to-video generation model.
- Section **B** provides more implementation details.
- Section **C** describes the limitation and failure cases.
- Section **D** describes the evaluation metrics, including rule-based, MLLM-based, and user study evaluations.
- Section **E** explains details of the MLLM-based evaluation.
- Section **F** presents computational efficiency analysis.
- Section **G** presents additional analyses and visualization results.
- Section **H** clarifies the use of large language models for editorial assistance in preparing this manuscript.

## A    INTRODUCTION OF THE BASE TEXT-TO-VIDEO GENERATION MODEL

We adopt a transformer-based latent diffusion framework (Peebles & Xie, 2023) as the foundation of our T2V generation model, shown in Figure 8. A 3D-VAE (Kingma & Welling, 2013) is first used to encode videos from pixel space into a latent representation, on which we build a transformer-based video diffusion model. Prior approaches often rely on UNets or transformers augmented with a separate 1D temporal attention module, but such designs that decouple spatial and temporal modeling typically limit performance. To address this, we replace the 1D temporal attention with 3D self-attention, allowing the model to jointly capture spatiotemporal dependencies and generate coherent, high-quality videos. Furthermore, before each attention and feed-forward network (FFN) block, we map the timestep to a scale and apply RMSNorm to the spatiotemporal tokens.

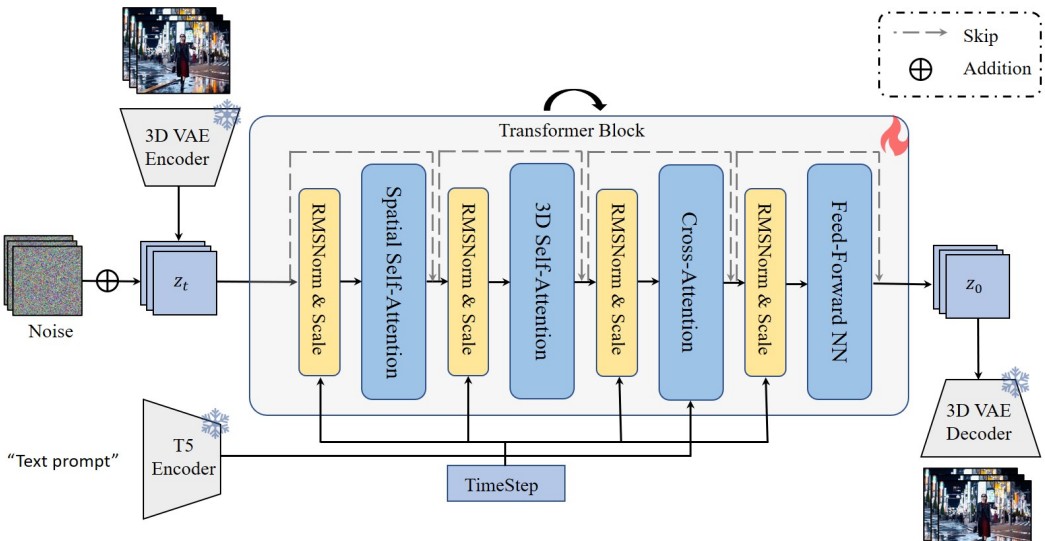

Figure 8: Overview of the base text-to-video generation model.

## B    IMPLEMENTATION DETAILS

During Stage I inference, we use 50 sampling steps with a timestep shift of 15, consistent with training. We inject only normalized 4D human motion without guided camera viewpoints. Since 4D

human motion mainly influences mid-to-high noise regions, we drop the motion condition in the last 10 steps to improve visual quality. The Stage I model uses CFG=5 for both text and human motion, as larger values tend to introduce artifacts. For Stage II inference, we also use 50 sampling steps with a timestep shift of 1, and no CFG is applied.

We use the GVHMR algorithm (Shen et al., 2024) to reconstruct 4D human motion data from videos. Moreover, GVHMR provides reconstructions in both camera and world coordinate systems, enabling us to compute the transformation $[R_{c2w} \mid T_{c2w}]$ and align the camera parameters with the human coordinate system. The focal length is set to the nominal values assumed by GVHMR, so our Stage II model only needs to estimate the extrinsic parameters. We transform the human data into a canonical space. Specifically, we define the gravity direction of the first frame as the negative $y$-axis, the frontal direction of the human body as the positive $z$-axis, and normalize the translation vector of the first frame to $0$. In this way, we obtain viewpoint-agnostic 4D human data.

## C    LIMITATION AND FAILURE CASES

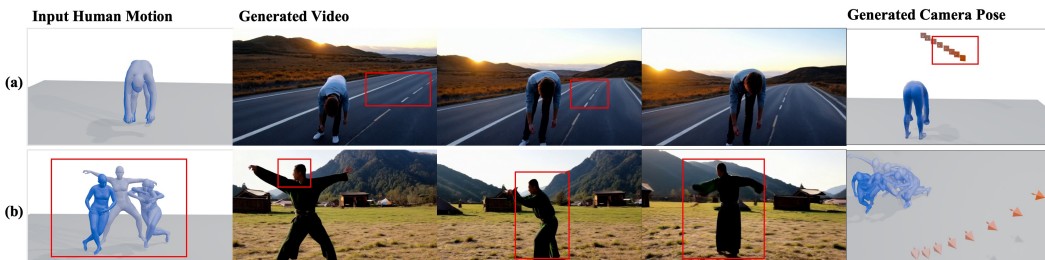

Figure 9: Visualization of failure cases. (a) indicates geometric distortions in the generated videos; (b) shows motion inconsistencies in complex action.

We identify that AdaViewPlanner has several limitations, which also highlight promising directions for future work.

First, for conceptual verification in this work, we simplify the 4D scene to a moving 3D human. This is based on the consideration that the human body is the core dynamic element in many scenes and can be conveniently represented using well-defined 3D models. While text instructions allow us to control scene-level content, the current method does not explicitly support more general 4D scenes or non-human dynamic entities. However, since our approach imposes no strict requirements on the 4D representation itself, it should be extendable to other non-human objects. In future work, we plan to adopt more general 4D representations (e.g., 4D Gaussian splatting, mesh deformation fields, or dynamic point clouds) and advanced 4D encoders to handle explicit 4D scene dynamics. We believe that as the capabilities of foundation video models continue to improve, our method will remain scalable.

Second, our method also inherits some of the base video model's shortcomings. For example, when the video generated by the Stage I model exhibits severe geometric inconsistencies, it leads to a decrease in the accuracy of the camera parameters predicted by the Stage II model, as shown in Fig. 9 (a). Additionally, our method can exhibit reduced motion consistency when handling complex human motions (Fig. 9 (b)). The main challenge is that the base video model often struggles to generate high-quality humans performing complex motions. We believe these issues can be mitigated by introducing geometric priors from 3D foundation models (e.g., VGGT (Wang et al., 2025)) to enhance the geometric consistency of the video model, as well as by using a more powerful base video model to better handle complex actions.

Third, the performance of our two-stage approach is intrinsically dependent on the adopted pretrained T2V model, as the major view planning knowledge is adapted from its learned representations. When the base video model has limited capability in this regard, our method can be significantly affected. Consequently, different base models may lead to different upper bounds on the achievable performance. Nevertheless, as video generation models continue to advance, our method is expected to benefit accordingly and achieve stronger results. We believe that our proof-of-concept provides non-trivial insights and inspiration for adapting T2V models to cinematic view planning.

## D    EVALUATION METRICS

Previous works, such as E.T., utilize metrics like Fréchet Camera Distance (FCD) and Text-Camera Consistency Score (CS). These are computed using a dedicated evaluation model, trained in a manner similar to CLIP (Radford et al., 2021) with camera and text encoders to learn a shared feature space. However, the performance of such an evaluator is closely tied to the quality and diversity of its training data. Given the current limitations in large-scale, diverse text-camera datasets, an evaluator trained on such data may exhibit a distributional bias. This can, in turn, affect the objectivity of the FCD and CS scores, particularly when assessing generations that fall outside the training distribution. Additionally, E.T. do not evaluate camera-human motion interaction. We therefore adapt and refine the reference-free metrics from DanceCamera3D for a more objective assessment.

**Rule-based Evaluation Metrics.** Following DanceCamera3D, we first adopt the Human Missing Rate (HMR), which measures whether the human is captured by the camera in each frame. Second, since high-quality trajectories are expected to be smooth and stable, we quantify camera smoothness by computing the jerk (the third derivative of motion) of both translational and rotational components; higher jerk values indicate greater shakiness. Finally, we refine the shot diversity metric in DanceCamera3D. Their approach evaluates the on-screen scale of the projected human model, which is view-invariant and thus unable to distinguish perspectives such as frontal vs. rear shots. To address this, we employ a geometric strategy: based on the character's visibility in each frame, we compute the camera's Euclidean distance and viewing angle relative to the character's orientation.

**MLLM-based evaluation.** Rule-based methods provide baseline evaluation but fail to capture higher-level aspects such as cinematographic style diversity and text-camera consistency. To address this, we employ advanced multimodal large models (e.g., Gemini 2.5 Pro) with task instructions and orthographic trajectory visualizations (top-down, front, side) to interpret camera motion relative to the character. For camera–text consistency, the model outputs a score from 0 (none) to 2 (perfect) with justification, while for cinematographic style, it evaluates perspective, distance, and motion type, using entropy to quantify diversity. Details are provided in the Appendix D.

**User study.** We invited 12 researchers in computer vision–related fields to participate in our user study. Their ages ranged from 21 to 30. All participants were informed of the purpose of the study and provided consent prior to participation. The study design and procedures were conducted in accordance with ethical standards to ensure the protection of participants' rights and privacy. Our user study consisted of two test sets: the E.T. test set with 21 samples and our own test set with 30 samples. In the questionnaire, we clearly explained the evaluation procedure and criteria to the participants. Specifically, each question presented the textual description of the camera motion along with three randomly ordered results generated by different methods. The evaluation criteria included: (1) consistency between the camera trajectory and the textual instruction, (2) professionalism of the camera motion, and (3) the coherence between the camera motion and the human actions. Each participant was asked to select the option they believed to be the best for each sample. We then aggregated all responses and computed the preference rate for each method on each test set. We provide an example of the user questionnaire in Table 13.

## E    DETAILS OF MLLM-BASED EVALUATION

Since there is currently a lack of automated metrics that can objectively evaluate the quality of trajectory generation from a high-level perspective, we propose leveraging advanced MLLMs (e.g., Gemini 2.5 Pro) to assess text–camera consistency and cinematographic style diversity. We observe that directly providing the model with a 2D video rendered from a 4D space containing both camera trajectories and human poses makes it difficult for the model to fully comprehend the underlying spatial relationships. To address this, we project the trajectories from three different viewpoints— top-down, front, and side—to obtain multi-view representations. We explicitly mark the start and end points of the trajectories, as well as their orientations, so that the model can incorporate this information when reasoning over the three-view inputs. Furthermore, we carefully design instruction templates (Tables 6 and 7) to guide the model toward producing accurate and relevant evaluations. Figure 10 illustrates several examples, with the rightmost column presenting the output results generated by Gemini 2.5 Pro.

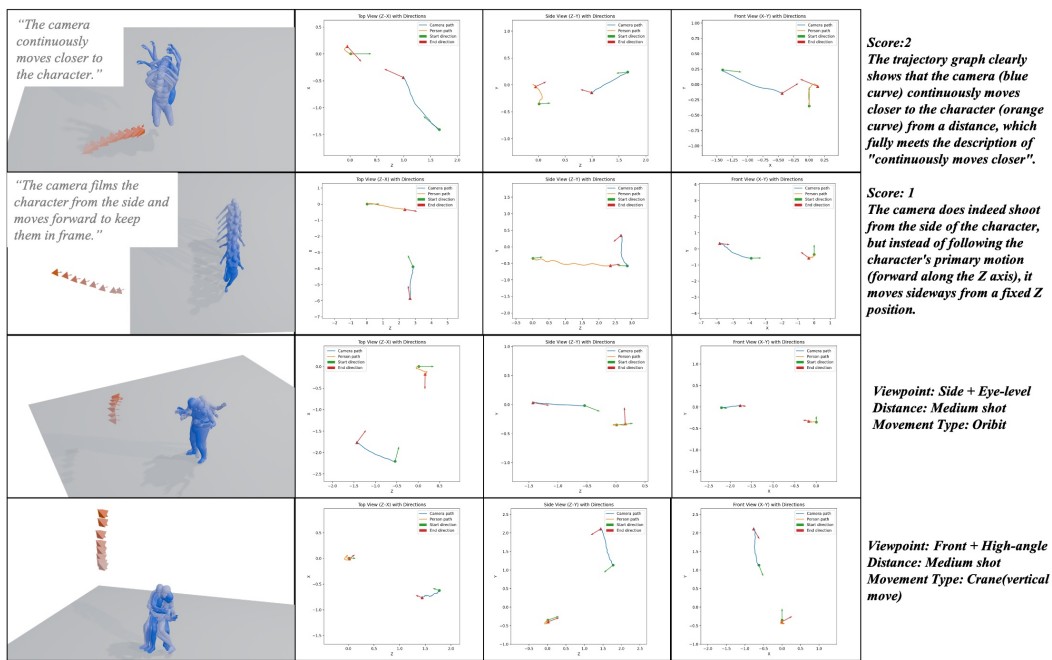

Figure 10: Examples of MLLM-based evaluation. Lines 1–2 assess text–camera consistency, while lines 3–4 evaluate the diversity of cinematographic styles. On the left, the textual input is paired with the corresponding 3D spatial visualization results. In the middle, the three-view projections are displayed, where both the tri-view images and the text serve as input data. On the right, the output generated by Gemini 2.5 Pro is presented.

To validate the reliability of this novel evaluation paradigm, we conducted a stability analysis by repeating the evaluation 10 times on the random 100 samples from Ours-240 test set. The results demonstrate the remarkable stability of our evaluation framework. Specifically, for the TCC metric, the mean evaluation score was 1.414, with a standard deviation (SD) of 0.082 and a coefficient of variation (CV) of 0.058. The CSD metric exhibited even greater stability, with a mean score of 0.646 (SD = 0.013, CV = 0.021). These exceptionally low variance metrics confirm that our MLLM-based evaluation is highly consistent and ensures the reproducibility of our reported results.

## F  COMPUTATIONAL EFFICIENCY ANALYSIS

We evaluate the computational efficiency of our framework, with detailed results on resource requirements and inference latency presented in Table 8. All experiments were conducted on a single NVIDIA A800 GPU, and the reported figures are an average of ten runs. The generated videos have a resolution of $672 \times 384$, and both the output video and the predicted camera parameters consist of 77 frames.

Table 9 further details the camera prediction performance across various inference step configurations. For the main results reported throughout this paper, we use a 50-step configuration for both stages to ensure optimal quality. Since our framework is based on video diffusion models, existing acceleration techniques (Lin et al., 2025; Lv et al., 2024) developed for diffusion models can be directly integrated to reduce inference latency.

## G  ADDITIONAL ANALYSES AND VISUALIZATION RESULTS

### G.1  COMPARISON WITH 3D CAMERA MOTION ESTIMATION

We highlight three key reasons for the necessity of training a Stage II model. First, existing approaches typically accept only videos or multiple images as input, without incorporating explicit

Table 6: Instruction Template for Text–Camera Consistency Evaluation

**Role**
You are a senior expert in virtual cinematography system evaluation, specializing in analyzing camera trajectory diagrams and assessing their alignment with text prompts.
**Objective**
Your core task is to evaluate the alignment between the **actual camera behavior** and the **camera behavior described in the text prompt**.
**Important Constraint:** Your evaluation must be based **only** on whether the camera's behavior (movement, rotation, relative position to the subject) matches the prompt. Do not evaluate whether the subject's actions match the prompt. Even if the subject's actions differ, as long as the camera follows the prompt's requirements in interacting with the subject, a positive evaluation should be given. Only assess the "camera–prompt" consistency.
**Input Format**
You will receive two core inputs:
**1. Triple-view Trajectory Images**
A set of static images showing the complete trajectories of both the camera and the subject.
- **Views included:** Top view (Z–X), Front view (X–Y), Side view (Z–Y)
- **Legend interpretation:**
- Start orientation: green arrows for initial orientation of camera/subject
- End orientation: red arrows for final orientation of camera/subject
- Subject trajectory: orange curve for subject's path
- Camera trajectory: blue curve for camera's path
**2. Text Prompt**
A description of the expected camera behavior. Examples:
- **Camera types:** push/pull, rotation, orbit, tracking, dolly, etc.
- **Camera–subject relation:** "Camera follows the character", "Camera orbits around the character".
**Core Evaluation Criteria**
1. **Camera Movement Type Consistency**
Does the camera's motion type match the description in the prompt?
*Example:* If the prompt says "orbit", does the camera circle around the subject? If the prompt says "follow", does the camera track the subject's movement?
2. **Camera–Subject Interaction Consistency**
Does the camera maintain the relationship or angle described in the prompt?
*Example:* If the prompt says "follow from behind", does the camera stay behind the subject? If the prompt says "zoom in during jump", does this happen correctly?
3. **Spatial Consistency**
Are the camera's direction, speed, and positioning logically consistent with the prompt?
*Example:* If the prompt says "rotate to the left", does the camera actually rotate left? If the prompt says "focus on the hand", does the camera's orientation reflect that?
**Key Judgment Principles**
1. The camera–subject viewpoint relationship should be inferred from the top view:
- Same orientation = camera behind subject
- Opposite orientation = camera in front
- Approximately perpendicular = camera on the side
2. Distance definition: close-up ($< 2\,\mathrm{m}$), medium shot ($2$–$4\,\mathrm{m}$), long shot ($> 4\,\mathrm{m}$).
3. Angle definition:
- Eye-level: height difference less than 0.5m
- High-angle: camera more than 1m above subject and facing negative y-axis direction
- Low-angle: camera more than 0.3m below subject and facing positive y-axis direction
**Scoring Rubric**
- 0 points: Camera behavior completely inconsistent with the prompt
- 1 point: Some aspects match, but others are missing or weakly related
- 2 points: Camera behavior fully matches the prompt in motion type, timing, and intention
**Output Format**
1. A single integer score: `0`, `1`, or `2`
2. One concise sentence summarizing the reason for the score

human motion conditions, which often leads to scale inconsistencies. Second, conventional camera estimation methods usually recover only relative and normalized trajectories; aligning them with the reference human pose coordinate system requires complex and computationally expensive post-processing. Third, AI-generated videos often exhibit mismatches in geometry and texture, which

Table 7: Instruction Template for Cinematographic Style Diversity

**Role**
You are a senior expert in virtual cinematography system evaluation, specializing in analyzing camera trajectory diagrams and identifying the type and characteristics of camera movements.
**Objective**
Your core task is to analyze the **type of camera movement**. You only need to output the movement categories, without explaining your reasoning process.
**Input Format**
**Triple-view Trajectory Images**
A set of static images showing the complete trajectories of both the camera and the subject.
    **Views included:** Top view (Z–X), Front view (X–Y), Side view (Z–Y)
    **Legend interpretation:**
       **Start orientation:** Green arrows represent the initial orientation of the camera/subject.
       **End orientation:** Red arrows represent the final orientation of the camera/subject.
       **Subject trajectory:** Orange curve represents the subject's path.
       **Camera trajectory:** Blue curve represents the camera's path.
**Classification Method**
Summarize camera movements from three perspectives:
1. **Viewpoint:** front, side, back, low-angle, high-angle, eye-level. The first three and the last three can be combined to form a viewpoint description.
2. **Shot scale:** close-up ($< 2\,\mathrm{m}$), medium shot ($2$–$4\,\mathrm{m}$), long shot ($> 4\,\mathrm{m}$).
3. **Movement type:** push-in, pull-out, orbit, static, rotation, tracking (horizontal move), crane (vertical move).
4. **Viewpoint definitions:**
- Eye-level: height difference less than 0.5m.
- High-angle: camera more than 1m above the subject and oriented downward.
- Low-angle: camera more than 0.3m below the subject and oriented upward.
**Output Format**
Your output must contain only the following three lines, strictly in this format, with each item on its own line. Do not include any other explanations, titles, introductions, conclusions, or punctuation such as semicolons. Do not add extra spaces after the colon.
**Format template:**
Viewpoint:[viewpoint classification]
Distance:[distance classification]
Movement type:[movement type classification]

**Valid output example:**
Viewpoint:Front+High-angle
Distance:Close-up
Movement type:Pull-out

Table 8: Inference time and memory usage under different step settings. The reported results are an average of ten runs a single NVIDIA A800 GPU.

| Setting | Stage I Time (s) ↓ | Stage II Time (s) ↓ | Stage I Memory (MB) | Stage II Memory (MB) |
|---|---|---|---|---|
| 50 steps | 31.60 | 16.20 | 15504 | 27290 |
| 25 steps | 15.99 | 8.41 | 15504 | 27290 |
| 10 steps | 6.39 | 3.66 | 15504 | 27290 |

significantly degrade the performance of feature-matching-based estimation algorithms (Li et al., 2025b; Zhang et al., 2024). This results in trajectory jitter, fragmented camera paths, and failures in scene reconstruction, as illustrated in Figure. 11.

## G.2    EVALUATION ON OPEN-SOURCE T2V MODEL

Here, we present the results obtained using the open-source T2V model, Wan2.2-5B (Wan et al., 2025). Specifically, Table 10 details the performance on 4D human motion control, while Table 11 reports the results for viewpoint planning. The evaluation demonstrates that Ours (Wan-2.2-5B) achieves performance second only to our original setting. In the future, we plan to further enhance

Table 9: Quantitative results under different inference step settings.

| Stage I & Stage II Steps | Rule-based Metrics | | | | | Reproject Acc | |
|---|---|---|---|---|---|---|---|
| | HMR ↓ | Jerk_t ↓ | Jerk_r ↓ | Dist_t ↑ | Dist_r ↑ | MSE ↓ | IoU ↑ |
| Stage I 50step & Stage II 50step | **0.018** | **0.003** | **0.001** | 1.415 | 0.529 | **0.158** | 0.338 |
| Stage I 50step & Stage II 25step | 0.019 | 0.003 | 0.001 | **1.416** | 0.528 | 0.164 | **0.340** |
| Stage I 50step & Stage II 10step | 0.021 | 0.004 | 0.002 | 1.388 | 0.517 | 0.165 | 0.329 |
| Stage I 25step & Stage II 50step | 0.026 | 0.003 | 0.001 | 1.394 | 0.527 | 0.188 | 0.277 |
| Stage I 25step & Stage II 25step | 0.026 | 0.003 | 0.001 | 1.387 | 0.525 | 0.188 | 0.279 |
| Stage I 10step & Stage II 50step | 0.025 | 0.003 | 0.001 | 1.347 | **0.536** | 0.195 | 0.273 |
| Stage I 10step & Stage II 10step | 0.027 | 0.004 | 0.002 | 1.349 | 0.534 | 0.194 | 0.269 |

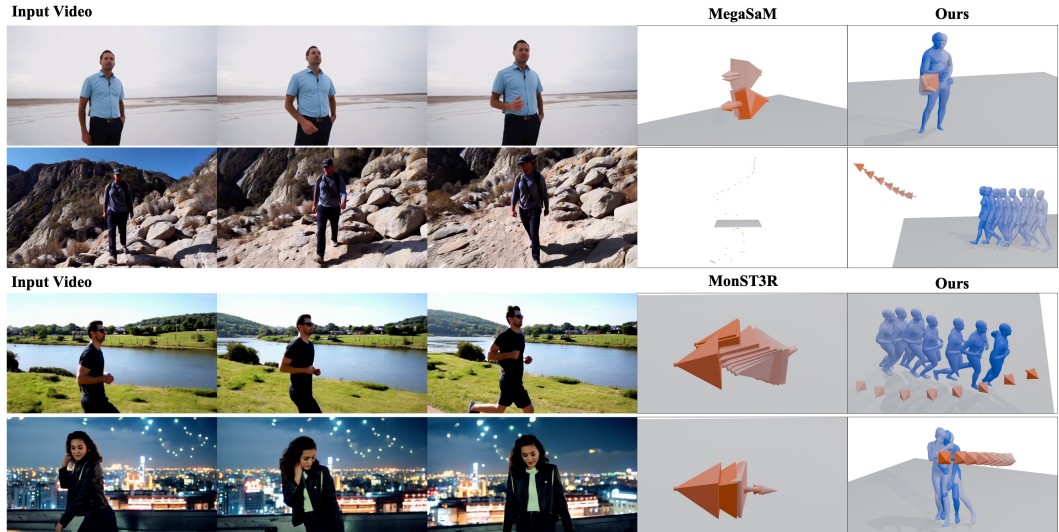

Figure 11: Comparison of camera estimation on AI-generated videos

Table 10: Quantitative comparison for 4D human motion control on TikTok (dance domain) and our curated general domain testsets. Here, *Ours (Wan-2.2-5B)* denotes our results trained on the open-source Wan-2.2-5B model (Wan et al., 2025).

| Method | TikTok (Dance Domain) | | General Domain | |
|---|---|---|---|---|
| | WA-MPJPE ↓ | PA-MPJPE ↓ | WA-MPJPE ↓ | PA-MPJPE ↓ |
| MTVCrafter (CogVideoX-5B) | 84.89 | 22.01 | 222.50 | 38.90 |
| MTVCrafter (Wan-2.1-14B) | 73.47 | **20.22** | 224.50 | 40.21 |
| Ours (Internal model) | **71.65** | 23.76 | **103.92** | **35.70** |
| Ours (Wan-2.2-5B) | 85.42 | 27.65 | 137.22 | 37.85 |

performance through more in-depth experimental exploration and by leveraging more advanced open-source video models.

### G.3 ABLATIONS ON MOTION FEATURES INJECTING MECHANISM

To investigate the effectiveness of our proposed motion feature injection mechanism, we conduct ablation studies on three alternative designs, with the results presented in Table 12. The designs are as follows: 1)MMDiT-style, which utilizes separate branches for video and motion features, and concatenates their respective tokens for joint spatial attention computation; 2)CrossDiT-style, where human motion tokens are injected as the key and value matrices into the spatial attention module; and 3) 3D Motion Attention, which replaces our Spatial Motion Attention module with a full 3D attention mechanism. Our analysis leads to the following observations.

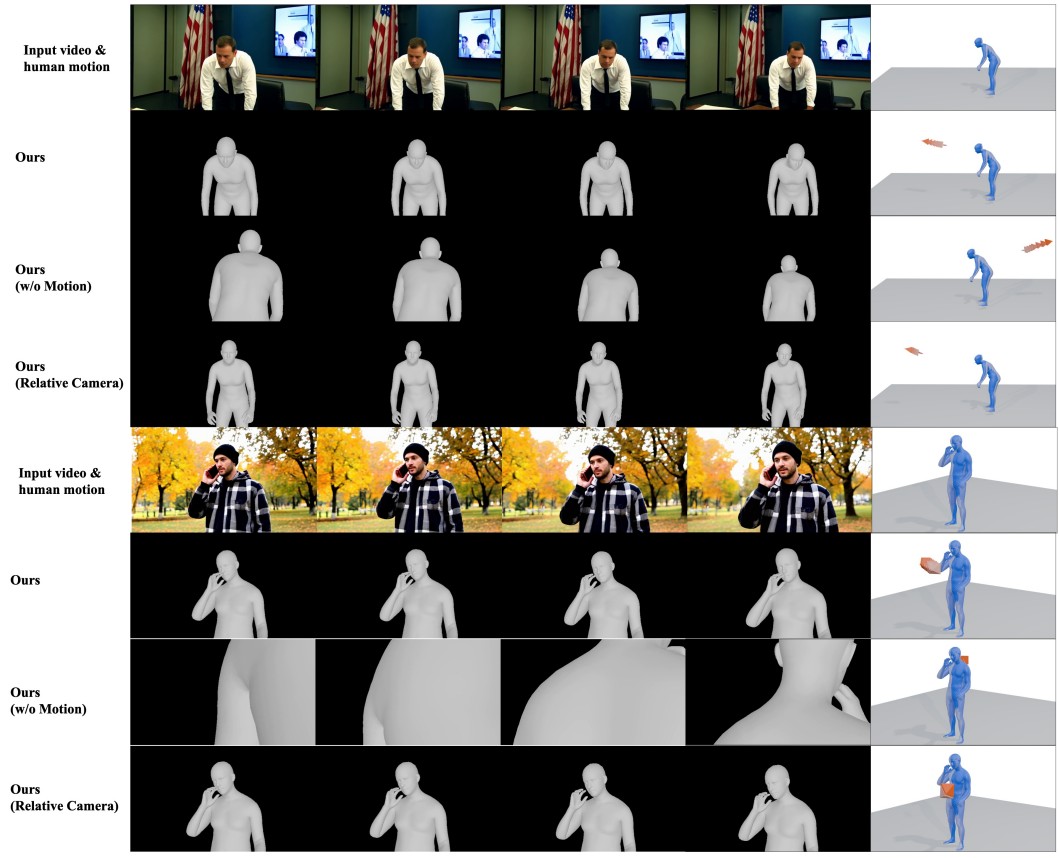

Figure 12: Additional ablation results on pipeline design.

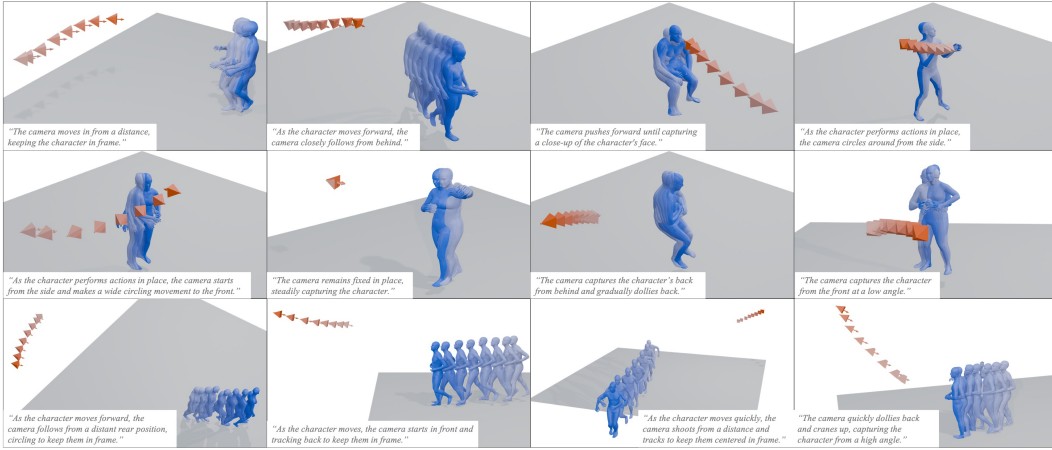

Figure 13: More visualization results of AdaViewPlanner, demonstrating the diversity of the generated trajectories and the model's ability to follow camera text instructions.

First, the **MMDiT-style** approach achieves comparable performance to our method within the same training duration. However, it introduces larger number of trainable parameters, suggesting that it may require extended training to converge to its optimal state. Second, the **CrossDiT-style** variant exhibits inferior performance. We hypothesize that this is because concatenating motion tokens facilitates a more comprehensive attention computation, thereby enabling the model to better capture the intricacies of 3D human motion. This concatenation-based fusion strategy is also adopted by

Table 11: Quantitative results on the E.T. testset and our curated testset. Here, *Ours (Wan-2.2-5B)* denotes our results trained on the open-source Wan-2.2-5B model ([Wan et al., 2025](#)).

| Method | Rule-based | | | | | MLLM-based | |
|---|---|---|---|---|---|---|---|
| | HMR $\downarrow$ | $\text{Jerk}_t \downarrow$ | $\text{Jerk}_r \downarrow$ | $\text{Dist}_t \uparrow$ | $\text{Dist}_r \uparrow$ | TCC $\uparrow$ | CSD $\uparrow$ |
| *E.T. Testset* | | | | | | | |
| E.T. | 0.064 | **0.001** | 0.026 | 0.538 | 0.540 | 0.850 | 0.608 |
| DanceCam* | 0.053 | 0.013 | 0.003 | 1.236 | 0.290 | 0.975 | 0.569 |
| Ours (Internal model) | **0.044** | 0.007 | **0.002** | **2.826** | **0.533** | 1.125 | **0.686** |
| Ours (Wan-2.2-5B) | 0.071 | 0.004 | **0.002** | 1.882 | 0.494 | **1.144** | 0.626 |
| *Ours Testset* | | | | | | | |
| E.T. | 0.048 | 0.001 | 0.029 | 0.700 | 0.225 | 0.790 | 0.623 |
| DanceCam* | 0.024 | 0.014 | 0.002 | 1.535 | 0.189 | 0.867 | 0.593 |
| Ours (Internal model) | **0.018** | **0.003** | **0.001** | **1.415** | 0.529 | **1.385** | **0.711** |
| Ours (Wan-2.2-5B) | 0.034 | **0.003** | **0.001** | 1.359 | **0.534** | 1.349 | 0.681 |

Table 12: Ablations on motion features injecting mechanism.

| Method | TikTok (Dance Domain) | | General Domain | |
|---|---|---|---|---|
| | WA-MPJPE $\downarrow$ | PA-MPJPE $\downarrow$ | WA-MPJPE $\downarrow$ | PA-MPJPE $\downarrow$ |
| MMDiT-style | 89.69 | 27.72 | 134.12 | 38.27 |
| CrossDiT-style | 127.02 | 32.07 | 206.84 | 41.62 |
| 3D Motion Attention | 131.88 | 31.88 | 192.36 | 39.66 |
| Ours | **71.65** | **23.76** | **103.92** | **35.70** |

prior works ([Fu et al., 2024](#)). Finally, the **3D Motion Attention** design compels the model to implicitly learn the correspondence between video tokens and motion tokens across different frames. This design significantly increases the learning difficulty. Given that the frame-wise correspondence between motion and video is explicitly known in our problem setting, our proposed Spatial Motion Attention is a more direct and effective design.

### G.4 MORE VISUALIZATION RESULTS

Figure. [12](#) presents additional ablation results on pipeline design, validating the rationality of our method. Fig. [13](#) shows the plausibility, diversity, and instruction consistency of the generated trajectories. Figure. [14](#) and [15](#) showcases further complete results, demonstrating the advanced capability of our approach in generating cinematic, diverse, and high-quality camera trajectories.

## H USE OF LARGE LANGUAGE MODELS

During the preparation of this manuscript, we made use of advanced language models (e.g., GPT-5, OpenAI, 2025) exclusively for editorial assistance. Their involvement was restricted to enhancing wording, improving clarity, and harmonizing style across sections. They were not used for generating research questions, designing methodologies, interpreting results, or drawing conclusions. All core ideas, experimental designs, and technical contributions are solely those of the authors. Moreover, every sentence edited with model support was carefully reviewed and approved by the human co-authors.

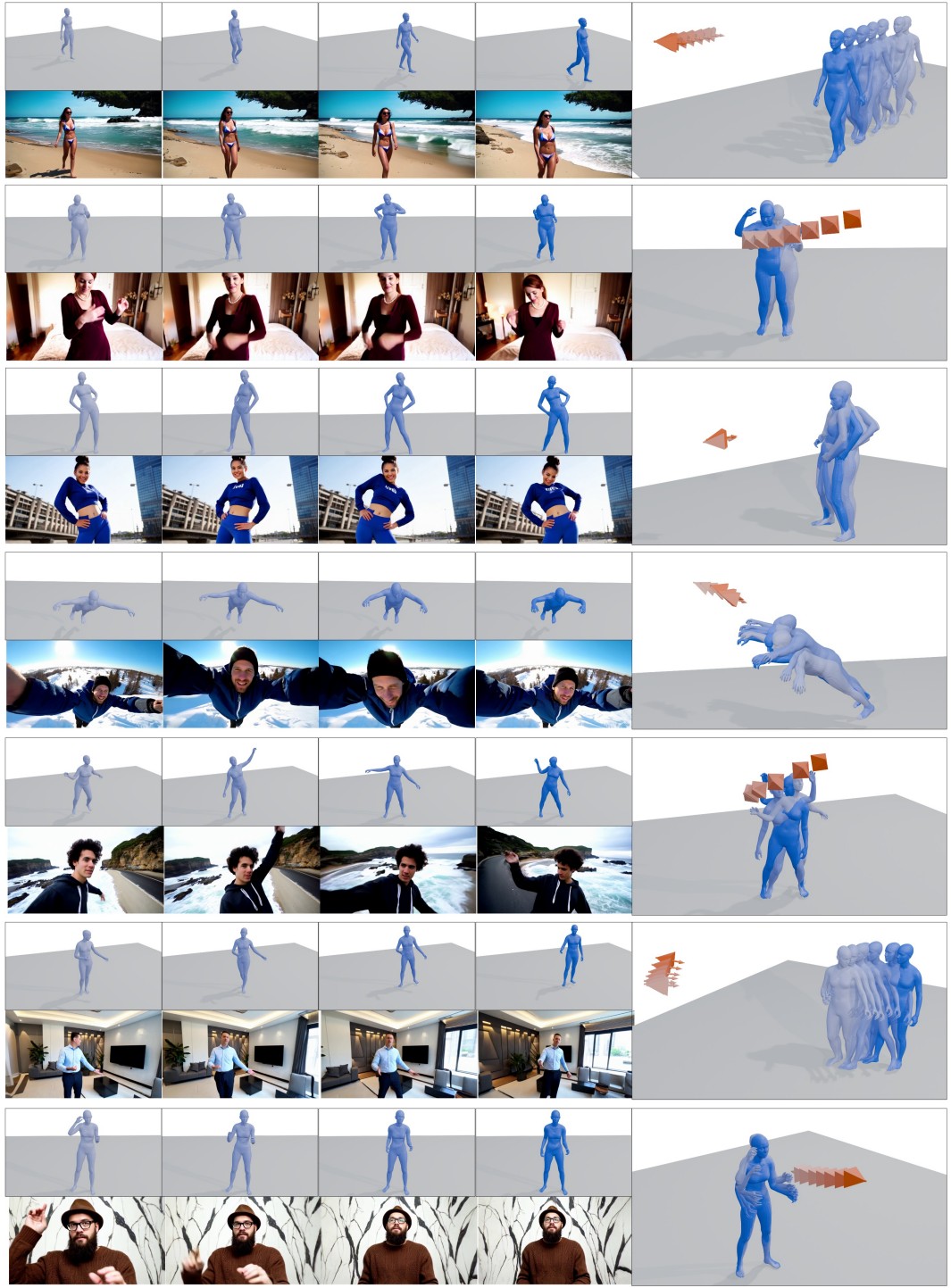

Figure 14: More visualization results

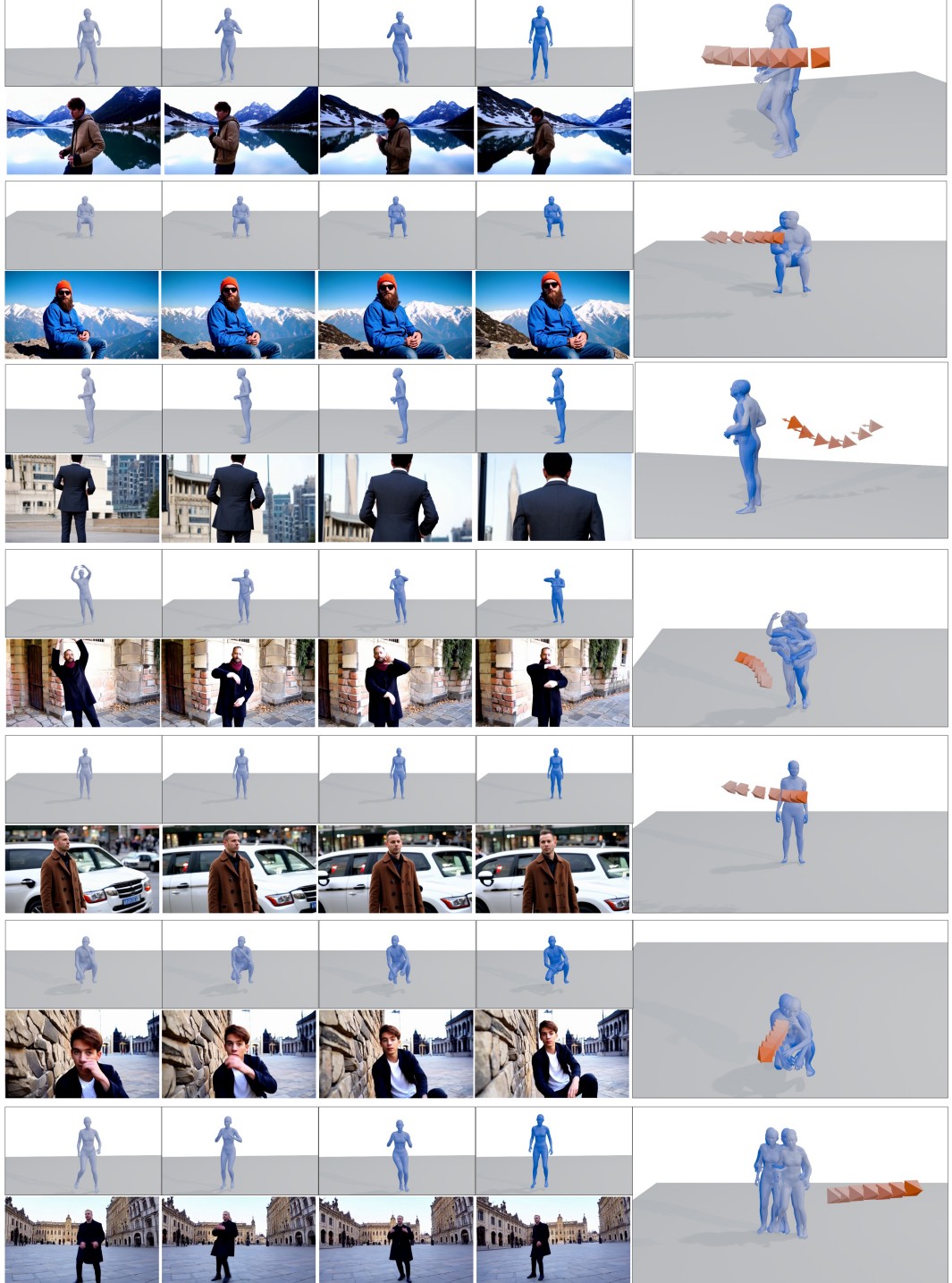

Figure 15: More visualization results

Table 13: User Questionnaire Example

**USER STUDY: A COMPARATIVE EVALUATION OF TEXT-DRIVEN CAMERA TRAJECTORY GENERATION FOR 4D SCENE**

### 1. Introduction and Informed Consent

Thank you for your interest in our study. This research aims to evaluate the performance of different AI models designed to generate camera trajectories for 4D scenes based on textual descriptions.

**Your participation in this study is completely voluntary.** You may withdraw at any time, for any reason, without penalty. The survey is expected to take approximately (e.g., 20–25 minutes) to complete.

**All responses collected will be fully anonymous**. We will not record any personally identifiable information. The aggregated, anonymized data will be used for academic research purposes only and may be published in a scientific paper. This study involves observing and evaluating short video clips and poses no anticipated risks.

By clicking "Proceed" to start the survey, you confirm that:

- You are 18 years of age or older.
- You have read and understood the information above.
- You voluntarily agree to participate in this study.

### 2. Task Description

In this study, you will be presented with a series of tasks. For each task, the goal is to evaluate an automatically generated camera trajectory based on a continuous 3D human motion and a text prompt describing the desired camera movement.

In each question, you will be presented with:

- **A Text Prompt:** A short sentence describing a specific type of camera movement (e.g., "A close-up shot focusing on the character's face," or "A dolly shot moving backward as the character moves").
- **Three Video Results:** Three short, auto-playing video clips labeled as Video A, Video B, and Video C. These are generated by different methods and their order is randomized for each question.

In the videos:

- The **gray character model** represents the predefined human action.
- The **red wireframe box** represents the camera's view frustum, visualizing the generated camera trajectory and field of view over time.

Your task: For each question, please watch the three videos and select the one you believe is the **best** result based on the evaluation criteria outlined below.

### 3. Evaluation Criteria

Please judge the results based on the following three aspects:

- **Consistency with Text Prompt:** How well does the generated camera trajectory match the textual description? Does it accurately perform the requested action (e.g., zoom, pan, follow)?
- **Professionalism & Cinematic Quality:** Does the camera movement appear professional, smooth, and visually appealing, as one might expect in a film?
- **Coordination with Human Action:** Is the camera movement well-coordinated with the character's actions? Does it effectively frame the character, highlight key moments, and create a coherent and engaging viewing experience?

### 4. Sample Question Illustration

**Question 1 / 51**:

**Text Prompt:** "An orbit shot around the character, starting from the front while keeping them centered."

**Results:** Video A, Video B, Video C.

**Question:** "Which method produced the best result?"

**Options:**

- Option 1: Video A is the best.
- Option 2: Video B is the best.
- Option 3: Video C is the best.

