# OpenReview forum: "AdaViewPlanner: Adapting Video Diffusion Models for Viewpoint Planning in 4D Scenes"
_ICLR.cc/2026/Conference — ICLR 2026 Poster_

### Official Review · Reviewer_paWE · 2025-10-20

**Soundness:** 2
**Presentation:** 3
**Contribution:** 2
**Rating:** 4
**Confidence:** 3

**Summary:**

The paper proposes AdaViewPlanner, a two-stage framework that adapts a pre-trained text-to-video (T2V) diffusion model to plan camera trajectories (viewpoints) for a given 4D scene, with a focus on human motion. Stage I injects normalized SMPL-X motion into a frozen T2V backbone via a spatial motion attention branch to generate cinematic videos whose frames implicitly encode the planned viewpoints; Stage II then extracts absolute camera extrinsics with a camera-diffusion branch in an MMDiT multi-modal transformer, conditioned on the Stage-I video and the 4D motion, trained with a flow-matching objective.

**Strengths:**

1. Writing and presentation are clear, with well-motivated design choices and well-explained results.

2. The paper is technically solid and presents meaningful improvements both quantitatively and visually.

3. The experiments are thorough, covering ablations, human evaluation, and clear baselines.

4. The method is conceptually useful because it connects generative video modeling with geometric camera control.

**Weaknesses:**

1. The main limitation is the narrow evaluation scope: all experiments focus on human motion, leaving unclear how the approach generalizes to other 4D scenes or dynamic objects.

2. I don't fully get the 4D scene concept in this paper. The generated scenes/humans are limited in view coverage.

3. The method depends heavily on accurate motion reconstruction via GVHMR, which may not be reliable in more complex scenes.

4. Another issue is reproducibility—part of the evaluation relies on a proprietary model (Gemini 2.5 Pro), which makes results hard to verify independently.

5. compute cost and inference efficiency are not reported, an important factor given the two-stage setup.

**Questions:**

1. How well does AdaViewPlanner perform on non-human or multi-object motion?

2. How sensitive is it to errors in motion reconstruction or camera intrinsics?

3. What is the computational overhead of the full pipeline, and could it be reduced through distillation or pruning?

4. Would a simpler cross-attention head perform comparably to the MMDiT design?

---

> ### Author Response · Authors · 2025-11-21
> **Official Comment by Authors (Part 1/2)**
>
> We are deeply grateful for the reviewer’s thoughtful insights and thorough evaluation of our manuscript. Please find what below our detailed point-to-point responses to all the comments.
>
> ---
> `[Q] Narrow evaluation scope: all experiments focus on human motion.`
>
> We appreciate this constructive comment. We utilizing the human skeleton as the 4D scene representation to validate the T2V model's capability as a view planner, just because of the well-defined keypoint topology of human.
>
> We acknowledge that other dynamic objects—unlike human figures—lack effective, easy-to-obtain structural representations and corresponding tools for extraction from video data, which limits their inclusion in the current validation. Crucially, our method does not impose strict requirements on the 4D representation itself and should be applicable to other non-human objects once their respective standardized representations are developed.
>
> Ultimately, we highlight that the demonstrated proof-of-concept already offers non-trivial insight and inspiration to the community regarding the foundation of T2V based view planning.
>
> ---
> `[Q] Clarification on the 4D scene concept.`
>
> In our paper, the 4D scene refers to the input of our model, which is simplified as 3D human skeleton sequence. So, it has nothing to do with view coverage because the scene/human is not generated by our model but provided by users (in the scenario of CG workflow). If we have misunderstood any part of your question, please do not hesitate to send us additional comments. We greatly welcome any further discussion.
>
> ---
> `[Q] Motion reconstruction accuracy of GVHMR may not reliable in complex scenes.`
>
> Our method demonstrates robust tolerance to minor or local disturbances within the input skeleton sequences. This is because effective camera movement hinges on the global semantics of the full body motion, rather than subtle local pose discrepancies. Therefore, our approach exhibits good robustness against the reconstruction accuracy limitations inherent in models like GVHMR. While extreme cases resulting in complete skeleton failure would certainly degrade performance, such instances were rarely encountered during our experiments.
>
> ---
> `[Q] Sensitivity to camera intrinsics.`
>
> We explain that our method is not sensitive to camera intrinsics. As described in Appendix B in the main paper, since we use GVHMR to reconstruct the human coordinate system, the camera intrinsics are set to the nominal values defined by GVHMR, which can be regarded as fixed intrinsic parameters. Therefore, our second-stage model only needs to predict the camera extrinsic parameters.
>
> ---
> `[Q] Regarding the reproducibility of MLLM-based evaluation`
>
> Thanks for raising this important point. In Appendix D (Evaluation Metrics) and E (Details of MLLM-based Evaluation), we describe how we conduct the MLLM evaluation, and in Tables 6 and 7 we provide the complete evaluation instructions for the two metrics. We plan to release the full evaluation code in the future. We believe that with the evaluation instructions, test examples, and the released code, our results can be reliably reproduced. In addition, we include an extra statistical evaluation metric here: for the random 100 samples from Ours-240 test sets, we repeat the evaluation 10 times and compute the standard deviation and coefficient of variation to assess the stability of outputs from Gemini 2.5 Pro.
>
> \begin{array}{l|ccc}
> \hline
> \textbf{Metric} & \textbf{Eval Result} & \textbf{Standard Deviation} & \textbf{Coefficient of Variation} \newline
> \hline
> \textbf{TCC} & 1.414 & 0.082 & 0.058 \newline
> \textbf{CSD} & 0.646 & 0.013 & 0.021 \newline
> \hline
> \end{array}
>
> The experimental results show that our evaluation achieves low standard deviation and coefficient of variation, indicating high stability and ensuring reproducibility of the evaluation.

---

> ### Author Response · Authors · 2025-11-21
> **Official Comment by Authors (Part 2/2)**
>
> `[Q] Regarding the compute cost and inference efficiency`
>
> Thanks for the constructive suggestion. We have included our computational efficiency results in Appendix F of the revised paper. Here we provide the memory usage, the inference time, and the corresponding evaluation metrics. The generated video resolution is 672×384, and both the generated video and the predicted camera parameters contain 77 frames. We evaluate the inference time and resource requirements ten times on a single **NVIDIA A800** and report the averaged results.
>
> \begin{array}{l|cccc}
> \hline
> \textbf{Infer Step} & \textbf{Stage I Time (s)} & \textbf{Stage II Time (s)} & \textbf{Stage I Memory (MB)} & \textbf{Stage II Memory (MB)} \newline
> \hline
> \text{50 step} & 31.60 & 16.20 & 15504 & 27290 \newline
> \text{25 step} & 15.99 & 8.41 & 15504 & 27290 \newline
> \text{10 step} & 6.39 & 3.66 & 15504 & 27290 \newline
> \hline
> \end{array}
>
> \begin{array}{l|ccccc|cc}
> \hline
> \textbf{Setting} & \textbf{Rule-based} & & & & & \textbf{Reproject Acc} & \newline
> \hline
> \text{Stage I and Stage II Steps} & \text{HMR} \downarrow & \text{Jerkₜ} \downarrow & \text{Jerkᵣ} \downarrow & \text{Distₜ} \uparrow & \text{Distᵣ} \uparrow & \text{MSE} \downarrow & \text{IoU} \uparrow \newline
> \hline
> \text{Stage I 50step and Stage II 50step} & \textbf{0.018} &\textbf{ 0.003} & \textbf{0.001} & 1.415 & 0.529 & \textbf{0.158} & 0.338 \newline
> \text{Stage I 50step and Stage II 25step} & 0.019 & 0.003 & 0.001 & \textbf{1.416} & 0.528 & 0.164 & \textbf{0.340} \newline
> \text{Stage I 50step and Stage II 10step} & 0.021 & 0.004 & 0.002 & 1.388 & 0.517 & 0.165 & 0.329 \newline
> \text{Stage I 25step and Stage II 50step} & 0.026 & 0.003 & 0.001 & 1.394 & 0.527 & 0.188 & 0.277 \newline
> \text{Stage I 25step and Stage II 25step} & 0.026 & 0.003 & 0.001 & 1.387 & 0.525 & 0.188 & 0.279 \newline
> \text{Stage I 10step and Stage II 50step} & 0.025 & 0.003 & 0.001 & 1.347 & \textbf{0.536} & 0.195 & 0.273 \newline
> \text{Stage I 10step and Stage II 10step} & 0.027 & 0.004 & 0.002 & 1.349 & 0.534 & 0.194 & 0.269 \newline
> \hline
> \end{array}
>
> The final reported results use 50 inference steps for both stages.  Since the core of our approach remains a video diffusion model, we anticipate that existing distillation or pruning techniques [1, 2] developed for diffusion models can be directly applied to reduce the inference latency of our framework.
>
> [1] Yahia, Haitam Ben, et al. "Mobile video diffusion." arXiv preprint arXiv:2412.07583 (2024).
>
> [2] Fang, Gongfan, et al. "Tinyfusion: Diffusion transformers learned shallow." Proceedings of the Computer Vision and Pattern Recognition Conference. 2025.
>
> ---
> `[Q] Regarding on Cross-Dit design`
>
> Thanks for the constructive suggestion. The core motivation of our work is to leverage the prior capabilities of video models. A key advantage of the MMDiT design is that we can fully reuse the architecture and weights of the original video model, requiring only two additional branches for camera pose and human motion. This design better inherits video-model priors and offers strong scalability. If we were to adopt a cross-DiT design, our flow-matching objective would then model camera parameters rather than videos, which would require restructuring the entire architecture and training from scratch. Given our primary goal of reusing the base video model's architecture and weights, we found the MMDiT architecture to be a more direct and suitable choice for the second stage. We would be happy to discuss any further ideas you may have.

---

> > ### Comment · Reviewer_paWE · 2025-11-27
> > **Good rebuttal**
> >
> > The authors have done a great job answering my doubts and questions. The provided additional details are very informative and convinced me to raise my rating. I would like to increase my rating to 6.

---

> > > ### Author Response · Authors · 2025-11-28
> > >
> > > Dear Reviewer paWE,
> > >
> > > Thank you for your positive feedback and for acknowledging that our detailed response and new experimental results have resolved your main concerns. We are truly grateful for your willingness to raise the rating to 6. This is a great encouragement to us.
> > >
> > > We noticed that the rating in the system is currently still displayed as 4. We wanted to kindly bring this to your attention in case there was a potential system issue. Your support in adjusting the score would be immensely appreciated.
> > >
> > > Thank you once again for your time, support, and significant contribution to improving our work.
> > >
> > > Best regards,
> > >
> > > Authors of paper #2886

---

### Official Review · Reviewer_HbVd · 2025-10-28

**Soundness:** 3
**Presentation:** 3
**Contribution:** 3
**Rating:** 6
**Confidence:** 4

**Summary:**

AdaViewPlanner adapts pre-trained text-to-video diffusion models to the task of viewpoint planning in 4D scenes by a two-stage method: (1) an adaptive learning branch injects viewpoint-agnostic 4D scene representations into the pre-trained T2V model so the generated conditional video visually encodes candidate viewpoints, and (2) a camera-extrinsic diffusion branch performs hybrid-condition guided denoising to extract camera extrinsics from the generated video and the 4D scene. The paper reports that this approach outperforms existing competitors and includes ablations validating the main design choices

**Strengths:**

1. Clear novel idea - leverages strong priors in large pre-trained video diffusion models as implicit world models to support viewpoint planning, reframing viewpoint prediction as a video-conditioned denoising task.

2. Practical two-stage design - separating scene injection and extrinsic prediction makes the method compatible with fixed pre-trained T2V backbones, reducing the need for fully retraining large video generators.

3. Empirical support - experiments claim superiority over competitors and ablation studies that isolate the contributions of the adaptive branch and the extrinsic diffusion branch.

4. Broader implication - demonstrates a promising direction of reusing generative video priors for embodied perception and 4D interaction tasks beyond pure generation.

**Weaknesses:**

1. Dependence on pre-trained T2V quality - performance likely tied to how well the base video diffusion model captures geometry and viewpoint cues; limited discussion of failure modes when the generator hallucinate inconsistent geometry. Experimental sensitivity to the choice of pre-trained model appears underexplored.

2. Scalability and compute - adapting and running diffusion branches for viewpoint extraction may be computationally intensive for real-time or embedded planning; paper does not clearly quantify runtime or resource requirements for planning in practice.

3. Generalization to real-world 4D data - the paper summary does not specify the datasets used or robustness to real sensor noise, occlusions, and dynamic scene elements, leaving open questions about transfer from synthetic or curated benchmarks to real-world robotics settings.

**Questions:**

1. Which pre-trained T2V backbones were used, and how sensitive are results to that choice? Can you please provide results with open-source T2V models?

2. What datasets were used for training and evaluation, and how do results vary between synthetic and real-world 4D scenes? Can you please provide results with data that is widely available?

3. What are the runtime and memory requirements for viewpoint extraction per candidate or per scene, and can the method be optimized for real-time planning?

4. How does the method handle dynamic scene elements or moving objects in the 4D input, and does it distinguish viewpoint changes from object motion?

5. Are there common failure modes (e.g., hallucinated geometry, ambiguous viewpoints), and do authors have strategies to detect or mitigate them?

6. Can the approach be extended to plan multi-step camera trajectories (sequences of viewpoints) rather than single-viewpoint prediction?

---

> ### Author Response · Authors · 2025-11-21
> **Official Comment by Authors (Part 1/4)**
>
> We sincerely appreciate this reviewer's insightful feedback and careful review of our manuscript. Please find what below our detailed point-to-point responses to all the comments.
>
> ---
> `[Q] Regarding the choice and dependence of the pre-trained T2V backbone`
>
> Our current experiments are conducted using an internal video model, and we describe its architecture in detail in Appendix A of the revised manuscript.
>
> We agree with that the performance of our two-stage approach is intrinsically dependent on the adopted pre-trained T2V model, as the major view planning knowledge is adapted from its learned representations. Empirically, we have observed that our model performs robustly across most scenarios, with performance degradation primarily limited to extremely complex or unconventional motion conditions.
>
> Regarding the sensitivity analysis across different base models, such a study would require substantial computational resources that are infeasible to allocate within the current review timeline. We agree on the importance of this factor and will address the issue through an intuitive discussion in the limitations section (Appendix C in the revised manuscript). We also plan to release code built on open-source models in the future.
>
> ---
> `[Q] Regarding the failure modes and improvement strategies`
>
> We have added a dedicated limitation and failure cases section in the revised version of the paper (Appendix C). We highlight the following failure modes:
> 1. Due to the limitations of the base video model, when the Stage I generated videos exhibit severe inconsistent geometry, the accuracy of the camera parameters predicted in the Stage II can be affected.
> 2. Our method may experience reduced pose consistency when handling extremely complex human motions.
> 3. When the textual instruction provides unclear descriptions of camera viewpoints or when the model fails to properly interpret them, ambiguous viewpoints may occur.
>
> We believe that there are several promising directions to address this problem in the future:
> - **Incorporating 3D priors into video models.** Recently, 3D foundation models such as VGGT [1] have demonstrated the ability to obtain comprehensive geometric understanding from 2D images via 3D-aware features. Some works have started to explore leveraging the 3D priors from VGGT to enhance the geometry consistency of video models [2,3].
> - **Handling complex human motions.** We identify two main challenges here: 1) The pre-trained base video model often struggles to generate high-quality humans performing complex motions; 2) Complex motions tend to degrade the accuracy of the annotated data. Therefore, we believe that stronger base video models and higher-quality data in the future may help alleviate this issue.
> - **Using a dedicated camera-motion text annotation model** to label training data, thereby improving the video model’s understanding of camera motion instructions.
>
> [1]Wang, Jianyuan, et al. "Vggt: Visual geometry grounded transformer." Proceedings of the Computer Vision and Pattern Recognition Conference. 2025.
>
> [2]Dai, Yixiang, et al. "FantasyWorld: Geometry-Consistent World Modeling via Unified Video and 3D Prediction." arXiv preprint arXiv:2509.21657 (2025).
>
> [3]Wu, Haoyu, et al. "Geometry forcing: Marrying video diffusion and 3d representation for consistent world modeling." arXiv preprint arXiv:2507.07982 (2025).

---

> ### Author Response · Authors · 2025-11-21
> **Official Comment by Authors (Part 2/4)**
>
> `[Q] Evaluation on compuation efficiency`
>
> We have included our computational efficiency results in the Appendix F of the revised paper.  Here we provide the memory usage , the inference time, and the corresponding evaluation metrics. The generated video resolution is 672×384, and both the generated video and the predicted camera parameters contain 77 frames. We evaluate the inference time and resource requirements ten times on a single **NVIDIA A800** and report the averaged results.
>
> \begin{array}{l|cccc}
> \hline
> \textbf{Infer Step} & \textbf{Stage I Time (s)} & \textbf{Stage II Time (s)} & \textbf{Stage I Memory (MB)} & \textbf{Stage II Memory (MB)} \newline
> \hline
> \text{50 step} & 31.60 & 16.20 & 15504 & 27290 \newline
> \text{25 step} & 15.99 & 8.41 & 15504 & 27290 \newline
> \text{10 step} & 6.39 & 3.66 & 15504 & 27290 \newline
> \hline
> \end{array}
>
> \begin{array}{l|ccccc|cc}
> \hline
> \textbf{Setting} & \textbf{Rule-based} & & & & & \textbf{Reproject Acc} & \newline
> \hline
> \text{Stage I and Stage II Steps} & \text{HMR} \downarrow & \text{Jerkₜ} \downarrow & \text{Jerkᵣ} \downarrow & \text{Distₜ} \uparrow & \text{Distᵣ} \uparrow & \text{MSE} \downarrow & \text{IoU} \uparrow \newline
> \hline
> \text{Stage I 50step and Stage II 50step} & \textbf{0.018} &\textbf{ 0.003} & \textbf{0.001} & 1.415 & 0.529 & \textbf{0.158} & 0.338 \newline
> \text{Stage I 50step and Stage II 25step} & 0.019 & 0.003 & 0.001 & \textbf{1.416} & 0.528 & 0.164 & \textbf{0.340} \newline
> \text{Stage I 50step and Stage II 10step} & 0.021 & 0.004 & 0.002 & 1.388 & 0.517 & 0.165 & 0.329 \newline
> \text{Stage I 25step and Stage II 50step} & 0.026 & 0.003 & 0.001 & 1.394 & 0.527 & 0.188 & 0.277 \newline
> \text{Stage I 25step and Stage II 25step} & 0.026 & 0.003 & 0.001 & 1.387 & 0.525 & 0.188 & 0.279 \newline
> \text{Stage I 10step and Stage II 50step} & 0.025 & 0.003 & 0.001 & 1.347 & \textbf{0.536} & 0.195 & 0.273 \newline
> \text{Stage I 10step and Stage II 10step} & 0.027 & 0.004 & 0.002 & 1.349 & 0.534 & 0.194 & 0.269 \newline
> \hline
> \end{array}
>
> The final reported results use 50 inference steps for both stages.  Note that, this paper focuses to explore the feasibility of using a pre-trained T2V model for view planning, with less emphasis placed on efficiency metrics. However, since the core of our approach remains a video diffusion model, we anticipate that existing speed-up techniques [1,2] for diffusion models will be directly applicable to our method.
>
> [1]Lin, Shanchuan, et al. "Diffusion Adversarial Post-Training for One-Step Video Generation." Forty-second International Conference on Machine Learning.
>
> [2]Lv, Zhengyao, et al. "FasterCache: Training-Free Video Diffusion Model Acceleration with High Quality." The Thirteenth International Conference on Learning Representations.
>
> ---
> `[Q] Regarding real-time planning`
>
> We clarify that our method is built on video diffusion models. Achieving real-time video generation remains challenging for current models. However, we believe that with improvements in base model capabilities and advances in acceleration techniques, our method has the potential to approach real-time planning.
>
> ---
> `[Q] Regarding handling dynamic scene elements and distinguishing viewpoint changes from object motion`
>
> As stated in the Method section of the paper, we simplify a 4D scene into a sequence of 3D human motions. This design choice is motivated by the fact that humans bodys can be effectively represented using mature 3D human models. We represent the dynamic human as a temporally continuous SMPL-X joints sequence and use it as a conditioning signal for the video model. Our method supports using text descriptions to control scene-level content, but it does not currently support explicit non-human dynamic elements as input. In future work, we plan to adopt more general 4D representations (e.g., 4D Gaussian splatting) together with advanced 4D encoders to support a broader range of explicit 4D scene dynamics.
>
> We clarify that a 4D scene itself does not contain viewpoint information. Typically, the 4D content is transformed into a canonical coordinate system before being used as a conditioning input. Our task is to plan camera viewpoints for such canonicalized, viewpoint-agnostic 4D scene content. If we have misunderstood any part of your question, we welcome further discussion.
>
> ---
> `[Q] Regarding multi-step camera trajectory planning`
>
> We clarify that our method is indeed capable of predicting multi-step camera trajectories. Here we restate the definition of our task: given a sequence of 3D human motions  **M ∈ ℝ^(F×K×3)**,  where F is the number of frames and K is the number of joints, our goal is to plan camera viewpoints for this 4D scene by generating camera trajectory parameters  **C ∈ ℝ^(F×9)**,  which match the frame length **F** of the input motion sequence. Therefore, our method naturally supports multi-step camera trajectory prediction. If we have misinterpreted your question in any way, we welcome further discussion.

---

> ### Author Response · Authors · 2025-11-21
> **Official Comment by Authors (Part 3/4)**
>
> `[Q] Regarding real-world 4D data evaluation`
>
> We have included real-world data evaluation in the Section 4.4 of the revised paper.
>
> First, we clarify that our method is not targeted for robotics settings but is designed for Computer Graphics (CG) animation workflows. Specifically, once an artist has completed a 4D animation, our approach is employed to predict the virtual camera trajectory and viewpoints that are better aligned with professional cinematography principles.
>
> Therefore, for the purpose of evaluation on real-world data, our primary requirement is the collection of 4D skeleton sequences derived from these professional animation scenarios.
> In typical CG production pipelines, 3D human skeletons originate from two primary sources: they are either synthetically generated (e.g., by game engines or professional designers) or captured from real-world performances using motion capture systems. Therefore, we choose the **GTA-Human** dataset [1] (3D human data generated by the GTA-V game engine) and the **AMASS** dataset [2] (multiple 3D human datasets captured by optical marker-based systems) as our real-world evaluation benchmarks. We randomly sample 100 test instances from each dataset for evaluation. The 4D human motion control and viewpoint planning evaluation results on real-world datasets are shown in the following table:
>
> \begin{array}{l|cc|cc}
> \hline
> \textbf{Method} & \textbf{AMASS} & & \textbf{GTA-Human} \newline
> \hline
>  & \text{WA-MPJPE} \downarrow & \text{PA-MPJPE} \downarrow & \text{WA-MPJPE} \downarrow & \text{PA-MPJPE} \downarrow \newline
> \hline
> \text{MTVCrafter (CogVideoX-5B)} & 107.57 & 49.58 & 270.75 & 55.61 \newline
> \text{MTVCrafter (Wan-2.1-14B)} & 103.69 & 45.78 & 240.72 & 56.46 \newline
> \textbf{Ours} & \textbf{72.19} & \textbf{42.13} & \textbf{145.00} & \textbf{50.23} \newline
> \hline
> \end{array}
>
> \begin{array}{l|ccccc|cc}
> \hline
> \textbf{Method} & \textbf{Rule-based} & & & & & \textbf{MLLM-based} & \newline
> \hline
>  & \text{HMR} \downarrow & \text{Jerkₜ} \downarrow & \text{Jerkᵣ} \downarrow & \text{Distₜ} \uparrow & \text{Distᵣ} \uparrow & \text{TCC} \uparrow & \text{CSD} \uparrow \newline
> \hline
> \textbf{AMASS Testset} & & & & & & & \newline
> \text{E.T.} & 0.033 & \textbf{0.002} & 0.015 & 0.422 & 0.150 & 0.900 & 0.564 \newline
> \text{DanceCam*} & 0.031 & 0.018 & 0.006 & 0.503 & 0.129 & 0.950 & 0.556 \newline
> \textbf{Ours} & \textbf{0.015} & 0.003 & \textbf{0.001} & \textbf{1.437} & \textbf{0.577} & \textbf{1.220} & \textbf{0.710} \newline
> \hline
> \textbf{GTA-Human Testset} & & & & & & & \newline
> \text{E.T.} & 0.110 & 0.004 & 0.063 & 0.620 & 0.330 & 0.880 & 0.585 \newline
> \text{DanceCam*} & 0.048 & 0.021 & 0.007 & 0.810 & 0.288 & 0.980 & 0.614 \newline
> \textbf{Ours} & \textbf{0.039} & \textbf{0.004} & \textbf{0.001} & \textbf{1.556} & \textbf{0.675} & \textbf{1.160} & \textbf{0.729} \newline
> \hline
> \end{array}
>
> Experimental results show that our method transfers well to real 4D data and exhibits strong generalization capability.
>
> [1] Cai, Zhongang, et al. "Playing for 3d human recovery." IEEE Transactions on Pattern Analysis and Machine Intelligence (2024).
>
> [2] Mahmood, Naureen, et al. "AMASS: Archive of motion capture as surface shapes." Proceedings of the IEEE/CVF international conference on computer vision. 2019.

---

> ### Author Response · Authors · 2025-11-21
> **Official Comment by Authors (Part 4/4)**
>
> `[Q] Regarding the description of training and evaluation datasets`
>
> In the Experimental Results section of the main paper, we introduced the training and evaluation data used in our experiments. Here, we provide a more detailed explanation:
> -  **Stage-1 training data.**  This set consists of 400k unfiltered videos and 10k carefully curated high-quality videos. Although these videos come from internal sources, their characteristics are general: all videos contain human subjects, and most include noticeable camera motions. Such data can also be easily obtained from public datasets by first collecting data with camera motion and then filtering for videos containing humans (for example, HumanVid [1] and MiraData [2]).
> - **Stage-2 training data.**  This set includes 101k MultiCamVideo [3], 43k HumanVid UE [1], and 100k internal UE videos. These datasets all feature precise camera parameter annotations and contain videos with human subjects.
> - **Stage-1 evaluation data.**  Similar to prior work on human motion control [4], we use TikTok [5] as the evaluation dataset for the first stage. Since TikTok videos primarily contain dance motions, we additionally filter 100 more general and diverse human-motion videos from internal data for evaluation.
> - **Stage-2 evaluation data.**  We randomly select 500 paired human motions and camera-instruction samples from the E.T. [6] test set. Because some motions from E.T. exhibit low quality (e.g., discontinuities, distortions), we also filter 240 high-quality human-motion sequences from internal data for evaluation.
>
> The human motion used in the above evaluation data is obtained using some 3D human reconstruction algorithms.
>
> [1] Wang, Zhenzhi, et al. "Humanvid: Demystifying training data for camera-controllable human image animation." Advances in Neural Information Processing Systems 37 (2024): 20111-20131.
>
> [2] Ju, Xuan, et al. "Miradata: A large-scale video dataset with long durations and structured captions." Advances in Neural Information Processing Systems 37 (2024): 48955-48970.
>
> [3] Bai, Jianhong, et al. "Recammaster: Camera-controlled generative rendering from a single video." arXiv preprint arXiv:2503.11647 (2025).
>
> [4] Ding, Yanbo, et al. "MTVCrafter: 4D Motion Tokenization for Open-World Human Image Animation." arXiv preprint arXiv:2505.10238 (2025).
>
> [5] Jafarian, Yasamin, and Hyun Soo Park. "Learning high fidelity depths of dressed humans by watching social media dance videos." Proceedings of the IEEE/CVF Conference on Computer Vision and Pattern Recognition. 2021.
>
> [6] Courant, Robin, et al. "ET the Exceptional Trajectories: Text-to-camera-trajectory generation with character awareness." European Conference on Computer Vision. Cham: Springer Nature Switzerland, 2024.

---

> ### Author Response · Authors · 2025-12-03
> **Official Comment by Authors (provide results on open-source T2V models)**
>
> `[Q] Evaluation on open-source T2V models`
>
> Thanks for the constructive suggestion. We fully agree that providing results on open-source models and releasing our code will be beneficial for the community to reproduce and evaluate our method. Due to limited time and computational resources, we provide here the results obtained using the open-source T2V model **Wan2.2-5B** [1].
>
> \begin{array}{l|cc|cc}
> \hline
> \textbf{Method} & \textbf{TikTok(Dance Domain)} & & \textbf{General Domain} & \newline
> \hline
>  & \text{WA-MPJPE} \downarrow & \text{PA-MPJPE} \downarrow & \text{WA-MPJPE} \downarrow & \text{PA-MPJPE} \downarrow \newline
> \hline
> \text{MTVCrafter (CogVideoX-5B)} & 84.89 & 22.01 & 222.50 & 38.90 \newline
> \text{MTVCrafter (Wan-2.1-14B)} & 73.47 & \textbf{20.22} & 224.50 & 40.21 \newline
> \text{Ours (Internal model)} & \textbf{71.65} & 23.76 & \textbf{103.92} & \textbf{35.70} \newline
> \text{Ours (Wan-2.2-5B)} & 85.42 & 27.65 & 137.22 & 37.85 \newline
> \hline
> \end{array}
>
>
> \begin{array}{l|ccccc|cc}
> \hline
> \textbf{Method} & \textbf{Rule-based} & & & & & \textbf{MLLM-based} & \newline
> \hline
>  & \text{HMR} \downarrow & \text{Jerk}_t \downarrow & \text{Jerk}_r \downarrow & \text{Dist}_t \uparrow & \text{Dist}_r \uparrow & \text{TCC} \uparrow & \text{CSD} \uparrow \newline
> \hline
> \textbf{E.T. Testset} & & & & & & & \newline
> \text{E.T.} & 0.064 & \textbf{0.001} & 0.026 & 0.538 & 0.540 & 0.850 & 0.608 \newline
> \text{DanceCam*} & 0.053 & 0.013 & 0.003 & 1.236 & 0.290 & 0.975 & 0.569 \newline
> \text{Ours (Internal model)} & \textbf{0.044} & 0.007 & \textbf{0.002} & \textbf{2.826} & \textbf{0.533} & 1.125 & \textbf{0.686} \newline
> \text{Ours (Wan-2.2-5B)} & 0.071 & 0.004 & \textbf{0.002} & 1.882 & 0.494 & \textbf{1.144} & 0.626 \newline
> \hline
> \textbf{Ours Testset} & & & & & & & \newline
> \text{E.T.} & 0.048 & \textbf{0.001} & 0.029 & 0.700 & 0.225 & 0.790 & 0.623 \newline
> \text{DanceCam*} & 0.024 & 0.014 & 0.002 & 1.535 & 0.189 & 0.867 & 0.593 \newline
> \text{Ours (Internal model)} & \textbf{0.018} & 0.003 & \textbf{0.001} & \textbf{1.415} & 0.529 & \textbf{1.385} & \textbf{0.711} \newline
> \text{Ours (Wan-2.2-5B)} & 0.034 & 0.003 & \textbf{0.001} & 1.359 & \textbf{0.534} & 1.349 & 0.681 \newline
> \hline
> \end{array}
>
> The result shows that **Ours (Wan-2.2-5B)** achieves results second only to our original setting. In the future, we plan to further improve the performance of open-source models through more in-depth experimental exploration and by leveraging more advanced open-source video models.
>
> [1] Wan, Team, et al. "Wan: Open and advanced large-scale video generative models." arXiv preprint arXiv:2503.20314 (2025).

---

### Official Review · Reviewer_fB3M · 2025-11-01

**Soundness:** 2
**Presentation:** 2
**Contribution:** 2
**Rating:** 4
**Confidence:** 4

**Summary:**

- This paper introduces AdaViewPlanner, the first method that adapts pre-trained Text-to-Video diffusion models for automatic camera viewpoint planning in 4D scenes.

- This paper uses a two-stage pipeline: (1) inject 4D scene into the T2V model to generate a video embedding implicit camera viewpoints, (2) extract camera poses via a dedicated diffusion branch conditioned on the video and scene.

- This paper provides outputs both coordinate-aligned camera trajectories and a video visualization, enabling prompt-controlled cinematography without requiring task-specific training.

**Strengths:**

- This paper leverages pretrained T2V priors, inheriting cinematic knowledge and strong generalization to diverse scenes—unlike previous specialized models requiring narrow datasets.

- This paper presents text-controllable viewpoint planning, enabling users to specify camera style and motion via natural language prompts.

- This paper ensures stable and effective design, with guided pose hints and a hybrid denoising branch that prevent training collapse and produce accurate, scene-aligned camera paths.

**Weaknesses:**

- Looking at 0001.mp4 (1 full result), there seems to be a tendency that the model does not fully reflect the motion, and I believe this should be mentioned in the limitations section.

- Only Stage I and Stage II are presented as the ablation study, but you should also include a more detailed ablation study on newly introduced components, such as Spatial Motion Attention.

- The user study is very unclear. It states "Invite researchers," but it is not specified what kind of researchers were involved. It also does not explain what results were obtained from the survey, and no example of the questionnaire is provided. Furthermore, it is not stated whether approval from the Institutional Review Board (IRB) was obtained. While detailed information cannot be disclosed due to the double-blind policy, I believe that at least the basic principles should be followed.

- Would the authors be willing to add a limitations section? It is necessary to provide clear information about what the method can and cannot do.

**Questions:**

Mentioned in the weaknesses

**Details Of Ethics Concerns:**

Mentioned in the weaknesses (User study)

---

> ### Author Response · Authors · 2025-11-21
> **Official Comment by Authors (Part 1/2)**
>
> We are deeply grateful for the reviewer’s thoughtful insights and thorough evaluation of our manuscript. Please find what below our detailed point-to-point responses to all the comments of this reviewer. We hope our responses can address this reviewer's concerns.
>
> ---
> `[Q] Regarding missing limitations section`
>
> We appreciate the reviewer’s constructive comment regarding the limitations section. We have added a section on limitations and failure cases to Appendix C in the revised version of the paper. Here, we summarize the main limitations of our work:
> 1. We currently simplify the 4D scene into moving 3D human. This is based on the consideration that the human body is the core dynamic element in many scenes and can be conveniently represented using 3D models. Although we can control scene-level content through text instructions, our method currently does not explicitly support general 4D scenes. In future work, we plan to adopt more general 4D representations (e.g., 4D Gaussian splatting]) and advanced 4D encoders to handle explicit 4D scene dynamics.
> 2. Since our model is built upon a text-to-video base model, it inevitably inherits some of the base model’s shortcomings. For example, the generated videos may exhibit severe geometric inconsistencies or hallucinations, and may show noticeably reduced motion consistency when handling complex human motions. These issues can lead to a decrease in the accuracy of the camera parameters predicted by the Stage II model. The main challenge is that the base video model has limited geometric consistency and often struggles to generate high-quality human subjects performing complex motions.
>
> ---
> `[Q] Regarding motion inconsistency`
>
> We clarify that our method allows for minor local motion inconsistencies in the videos generated by Stage I, as their impact on the camera parameters predicted by Stage II is negligible. This is because effective camera movement depends primarily on the global semantics of the full-body motion rather than subtle local pose discrepancies. While extreme cases that result in complete skeletal failure would certainly degrade performance, such cases were rarely observed in our experiments.
>
> ---
> `[Q] Regarding ablations on Spatial Motion Attention component`
>
> We sincerely thank this reviewer for the insightful suggestion. We have added an ablation study on the Spatial Motion Attention component in Section 4.4 of the revised paper. We designed three ablations for injecting motion features into the model:
> 1. **MMDiT-style**, where the model contains separate video and motion branches, and motion tokens are concatenated with spatial tokens for joint spatial attention computation.
> 2.  **CrossDiT-style**, where human motion tokens are injected as keys and values into the spatial motion attention.
> 3. **3D Motion Attention**, where the Spatial Motion Attention module is replaced with a 3D Motion Attention mechanism.
>
> The ablation results are as follows:
>
> \begin{array}{l|cc|cc}
> \hline
> \textbf{Method} & \textbf{TikTok (Dance Domain)} & & \textbf{General Domain} & \newline \hline
> & \text{WA-MPJPE} \downarrow & \text{PA-MPJPE} \downarrow & \text{WA-MPJPE} \downarrow & \text{PA-MPJPE} \downarrow \newline
> \hline
> \text{MMDiT-style} & 89.69 & 27.72 & 134.12 & 38.27 \newline
> \text{CrossDiT-style} & 127.02 & 32.07 & 206.84 & 41.62 \newline
> \text{3D Motion Attention} & 131.88 & 31.88 & 192.36 & 39.66 \newline
> \textbf{Ours} & \textbf{71.65} & \textbf{23.76} & \textbf{103.92} & \textbf{35.70} \newline
> \hline
> \end{array}
>
> Our analysis is as follows:
>
> - **MMDiT-style:** Under the same training time, this approach achieves performance close to Spatial Motion Attention. However, because MMDiT introduces a larger number of trainable parameters, it likely requires longer training to reach optimal performance.
>
> - **CrossDiT-style:** This variant shows relatively poor results. We believe that concatenating motion tokens enables a more complete attention computation, allowing the model to better understand 3D human motions (e.g., prior works such as 3dtrajmaster[1] also adopt concatenation-based fusion).
>
> - **3D Motion Attention:** This design forces the model to infer which video tokens correspond to which motion tokens across frames, which significantly increases learning difficulty. Since the correspondence between human motion frames and video frames is known in our setting, using Spatial Motion Attention is the more reasonable design.
>
> [1] Fu, Xiao, et al. "3dtrajmaster: Mastering 3d trajectory for multi-entity motion in video generation." arXiv preprint arXiv:2412.07759 (2024).

---

> ### Author Response · Authors · 2025-11-21
> **Official Comment by Authors (Part 2/2)**
>
> `[Q] Regarding unclear user study`
>
> We appreciate your thoughtful feedback regarding the shortcomings in our user study description and the absence of an explicit ethics statement. We have revised the paper to provide a more detailed description of the user study (Section 4.1 and Appendix D), include an example questionnaire (Table 11), and improve the Ethics Statement (Page 11).
>
> 1）**Regarding the User Study Details:**
> - **Participant Overview.**  We invited 12 researchers in computer vision–related fields to participate in our user study. Their ages ranged from 21 to 30. All participants were informed of the purpose of the study and provided consent prior to participation. The study design and procedures were conducted in accordance with ethical standards to ensure the protection of participants’ rights and privacy.
>
> - **Evaluation Procedure.**  The user study consisted of two test sets: the E.T. test set with 21 samples and our own test set with 30 samples. In the questionnaire, we clearly explained the evaluation procedure and criteria to the participants. Specifically, each question presented the textual description of the camera motion along with three randomly ordered results generated by different methods.
>
> - **Evaluation Criteria.**  Participants were instructed to evaluate results based on:
>   (1) The consistency between the camera trajectory and the textual instruction.
>
>   (2) The professionalism of the camera motion.
>
>   (3) The coherence between the camera motion and the human actions.
>
> - **Computation Method.**  Each participant was asked to select the option they believed to be the best for each sample. We then aggregated all responses and computed the preference rate for each method on each test set.
>
> - **Results Analysis.**  The results (Table 1 in the paper) show that our method achieved over 60% user preference across both test sets, indicating that the camera trajectories generated for 4D scenes were consistently favored by the participants.
>
> 2）**Regarding the IRB Approval and Ethical Considerations:**
> We take research ethics very seriously and strictly followed the ICLR Code of Ethics during the study. Specifically, we implemented the following measures to protect participants:
> - **Informed consent:** Before the study, all participants were provided with a detailed informed consent form explaining the study procedures, and their explicit consent was obtained.
> - **Anonymization:** All collected data were anonymized to ensure the protection of participants’ rights and privacy.
> - **Voluntary participation:** We clearly informed participants that their participation was entirely voluntary, that they could withdraw at any time without providing a reason, and that their data would be deleted upon withdrawal.
> - **Minimal risk:** The study involved minimal risk, ensuring that participants were not exposed to any harm.
>
> Given the minimal-risk nature of the study and the protective measures adopted, we believe that our research does not require ethical approval from an Institutional Review Board. We have added these clarifications to the Ethics Statement in the revised version of the paper. We are committed to strictly adhering to the Code of Ethics in all future research.

---

> > ### Comment · Reviewer_fB3M · 2025-11-26
> >
> > Thank you for addressing and reviewing the various concerns raised. However, you stated, "Given the minimal-risk nature of the study and the protective measures adopted, we believe that our research does not require ethical approval from an Institutional Review Board." It is important to note that this is not a determination for the authors to make, but rather one that should be made by the IRB. Phrasing it this way may lead to misunderstanding, and I kindly ask that you keep this in mind moving forward. Taking your clarifications into account, I have removed the ethics flag and increased the rating by one level. Happy Thanksgiving.

---

> > > ### Author Response · Authors · 2025-11-26
> > >
> > > Dear Reviewer fB3M,
> > >
> > > Thank you very much for your valuable feedback and for raising our paper's rating. We sincerely appreciate your guidance.
> > >
> > > We fully accept your point regarding the IRB determination. We will revise the corresponding statement in the final version of our manuscript. Furthermore, we will strictly adhere to the Code of Ethics in all our future research.
> > >
> > > Thank you again for your constructive comments. Happy Thanksgiving!
> > >
> > > Best regards,
> > >
> > > Authors of paper #2886

---

### Official Review · Reviewer_rbXy · 2025-11-01

**Soundness:** 4
**Presentation:** 4
**Contribution:** 4
**Rating:** 8
**Confidence:** 3

**Summary:**

This paper proposes AdaViewPlanner, which leverages pre-trained Text-to-Video (T2V) models to automatically generate professional camera trajectories in 4D scenes. The core contribution is a two-stage pipeline: Stage I injects 4D human motion into a T2V model to generate cinematic videos with implicit camera movements using a guided learning scheme; Stage II explicitly extracts camera poses through a camera diffusion branch in an MMDiT framework. The method outperforms baselines on multiple metrics and demonstrates text-controllable, diverse camera trajectory generation.

**Strengths:**

1. Novel and well-motivated approach. First work to leverage pre-trained T2V models for automatic camera planning in 4D scenes, based on the insight that T2V models implicitly learn cinematographic knowledge. This approach effectively reuses foundation model priors for generalization and text-controllability.
2. Effective guided learning scheme. The curriculum learning strategy (providing ground-truth camera tokens with probability p) is critical for preventing training collapse. Ablations clearly show variants without this guidance or the video model fail to converge (Figure 7, Table 2).
3. Strong experimental results. Significant improvements over baselines across all metrics (>60% user preference, Table 1), with comprehensive evaluation addressing prior limitations. Thorough ablations validate each design choice.

**Weaknesses:**

1. Heavy reliance on synthetic data with limited real-world validation. Stage II training depends entirely on synthetic UE datasets (244k samples) and GVHMR reconstructions, with no evaluation on real captured videos. The sim-to-real transfer capability remains undemonstrated, and reconstruction errors from GVHMR directly propagate to training, potentially limiting real-world robustness.
2. Limited scope. Evaluation focuses solely on human-centric SMPL-X motion with no experiments on multi-agent scenes, non-human subjects, or general dynamic objects. (This can be future work)

**Questions:**

3D RoPE temporal sensitivity and motion speed dependency. Does the 3D RoPE encoding exhibit sensitivity issues across different motion speeds? Since actions vary from slow walking to fast dancing, have you analyzed whether the positional encoding properly captures these temporal dynamics? Table 2 shows modest improvement with 3D RoPE (122.13 vs 103.92 WA-MPJPE), but does performance degrade for extremely fast or slow motions? Have you examined whether motion-speed-adaptive encoding or temporal scaling strategies could improve results, particularly for rapid actions requiring finer temporal resolution versus slow motions needing different spatial emphasis?

---

> ### Author Response · Authors · 2025-11-21
>
> We sincerely appreciate this reviewer's insightful feedback and careful review of our manuscript. Please find what below our detailed point-to-point responses to all the comments of this reviewer. We hope our responses can address this reviewer's concerns.
>
> ---
> `[Q] Concerns about stage II training entirely on UE datasets and GVHMR reconstructions, without evaluation on real captured videos.`
>
> We would like to clarify that the stage II model never need to tackle real captured videos because its input would always be the AI-generated video by T2V model (at stage I). The majore rational of using UE dataset for stage II training is that UE data offers videos paired with ground-truth camera parameters. Although GVHMR may introduce some errors to estimated skeletons, it would not be a problem because the reasonable camera movement should hinge on the global semantics of the full body motion rather than subtle local pose discrepancies.
> Besides, extensive experiments validated that our proposed method achieves notable superiority over existing competitors.
>
> ---
> `[Q]  Limited scope: evaluation on human-centric motion only.`
>
> We appreciate this constructive comment. Our current focus is centered on utilizing the human skeleton as the 4D scene representation primarily to validate the T2V model's capability as a view planner, largely because the skeleton possesses a well-defined and standardized keypoint topology.
>
> We acknowledge that other dynamic objects—unlike human figures—lack effective, easy-to-obtain structural representations and corresponding tools for extraction from video data, which limits their inclusion in the current validation. Crucially, our method does not impose strict requirements on the 4D representation itself and should be applicable to other non-human objects once their respective standardized representations are developed.
>
> While this complexity limits their inclusion in the current validation, we believe our proof-of-concept offers non-trivial insights and inspiration to the community regarding the adaptation of T2V models for cinematic view planning.
>
> ---
> `[Q] Regarding discussion on 3D RoPE`
>
> We highly appreciate your suggestion regarding the motion-speed-adaptive encoding strategy, and we agree that it is a very valuable idea. First, we provide a detailed explanation of the 3D RoPE used in our method, which follows the design of MTVCrafter [1]. Specifically, the three dimensions in our 3D RoPE correspond to the spatial coordinates (x, y, z). These spatial statistics are obtained by computing the average position of each joint across the entire dataset. Based on these statistics, we compute sinusoidal RoPE features for each of the three dimensions and concatenate them to obtain the final RoPE feature. This design helps the model better understand 3D human motion in space.
>
> Since we inject motion features into the spatial motion attention module, we only need to compute RoPE in the spatial domain. At this design, we already know the alignment between each video token and its corresponding motion token. Therefore, the 3D RoPE we compute does not incorporate dynamic temporal information. As a result, the 3D RoPE encoding we use is not sensitive to motion speed.
>
> Nevertheless, the motion-speed-adaptive encoding strategy you propose — enabling the model to perceive changes in motion speed and better capture temporal dynamics — is both reasonable and insightful. We believe that incorporating such an adaptive encoding mechanism has the potential to further improve the effectiveness of 3D motion attention design. We plan to explore this idea in future work, and we sincerely appreciate your suggestion.
>
> [1] Ding, Yanbo, et al. "MTVCrafter: 4D Motion Tokenization for Open-World Human Image Animation." arXiv preprint arXiv:2505.10238 (2025).

---

### Author Response · Authors · 2025-11-22
**Global Reply**

Dear Reviewers,

We sincerely thank all reviewers for their thoughtful comments and constructive feedback.

We have carefully considered each point and provided clarifications and justifications accordingly. Detailed responses are included below. In addition, we have uploaded a **revised PDF** in which all modifications are **highlighted in blue**. Specifically, we have made the following updates:

- **Section 4.4**: Added evaluation results on real-world 4D data.
- **Appendix C**: Added a dedicated section on limitations and failure cases.
- **Appendix F**: Added a computational efficiency analysis.
- **Appendix G.2**: Added ablation studies on the motion-feature injection mechanism.
- **Section 4.1, Appendix D, and Table 11**: Provided additional user study details and a questionnaire example, and strengthened the Ethics Statement.
- **Appendix E**: Added a stability analysis for the MLLM-based evaluation.

We hope that our explanations adequately address all concerns. Please feel free to let us know if any further details or clarifications would be helpful.

Best regards,
Authors of Paper #2886

---

### Author Response · Authors · 2025-12-03
**Summary of Work and Rebuttal for Submission 2886**

Dear Area Chair,

Thank you for taking over the evaluation of our submission. Below, we provide a brief summary of our work and the rebuttal updates.

---
### **Work Summary**
Inspired by the powerful capabilities of recent text-to-video (T2V) models, our work aims to explore the feasibility of repurposing such models as virtual cinematographers to design professional camera trajectories for 4D scenes. Our key insight is that pre-trained T2V models can generate vivid dynamic content with professional camera movements based on text prompts, indicating their internal cinematographic knowledge and instruction-following capabilities. We propose a two-stage paradigm to adapt pre-trained T2V models for viewpoint prediction. Experimental results demonstrate the effectiveness of our proposed method and the soundness of our key technical designs.

Prior to the rebuttal, the work was recognized for its "novel and well-motivated approach" (rbXy, HbVd), "reasonable method design" (rbXy, fB3M, HbVd, paWE), and "strong experimental results" (rbXy, HbVd, paWE).

---
### **Key Concerns Raised by Reviewers and Our Responses**

1. **Common concerns**
    * **Absence of limitations and failure cases analysis.** We have added a dedicated section analyzing limitations and failure cases in the revised manuscript (App. C, Fig. 9).
    * **Absence of computational efficiency analysis.** We have added detailed results on memory usage, inference time, and corresponding evaluation metrics under different inference step settings (App. F, Tables 8 and 9). Since the core of our approach remains a video diffusion model, we anticipate that existing speed-up techniques for diffusion models will be directly applicable to our method.
    * **Regarding motion reconstruction error.** We explained that our method demonstrates robust tolerance to minor errors, as effective camera movement hinges on the global semantics of the full-body motion rather than subtle local pose discrepancies.

2. **rbXy (8, no post-rebuttal response)**
    * **Concerns about evaluation on real captured videos.** We clarified that the Stage II model never needs to tackle real captured videos because its input is always the AI-generated video from the Stage I T2V model.

3. **fB3M (4 $\to$ 6)**
    * **Ablations on the design of motion feature injection.** We designed three ablation schemes: MMDiT-style, CrossDiT-style, and 3D Motion Attention. We analyzed the characteristics of each scheme based on the results and demonstrated that the current Spatial Motion Attention design is optimal (App. G.3, Table 12).
    * **Unclear User Study.** We have revised the manuscript to provide a more detailed description of the user study (App. D), include an example questionnaire (Table 13), and improve the Ethics Statement (Page 11).

4. **HbVd (6, no post-rebuttal response)**
    * **The choice of the pre-trained T2V backbone.** We acknowledge that the performance of our two-stage approach is intrinsically dependent on the adopted pre-trained T2V model. Following the reviewer's suggestion, we provided results based on the open-source T2V model, Wan2.2-5B (App. G.2, Tables 10 and 11).
    * **Real-world 4D data evaluation.** We clarified that our method is designed for Computer Graphics (CG) animation workflows and explained our rationale for selecting the real-world test set. We provided results for all methods on these test sets and analyzed that our method transfers well to real 4D data, exhibiting strong generalization capability (Sec. 4.4, Tables 4 and 5).
    * **Potential strategies for addressing failure modes.** We discussed potential future strategies to address the current failure modes of the model.

5. **paWE (4 $\to$ 6)**
    * **Clarification on the 4D scene concept.** We clarified that "4D scene" refers to the input of our model, which is simplified as a 3D human skeleton sequence (Sec. 3).
    * **The reproducibility of MLLM-based evaluation.** In Appendices D and E, we described how we conduct the MLLM evaluation, and in Tables 6 and 7, we provided the complete evaluation instructions. In addition, we calculated the standard deviation and coefficient of variation to assess the stability of outputs from Gemini 2.5 Pro. The experimental results demonstrate that our evaluation possesses high stability.

---
### **Conclusion**
Our rebuttal provides detailed responses to the reviewers’ concerns. Reviewers fB3M and paWE have indicated that our response successfully addressed their concerns. All updates in the revised manuscript are highlighted in **blue**.

We hope these revisions clearly convey AdaViewPlanner’s contributions and benefits to the ICLR community, and we appreciate your time in evaluating our work.

---
Best regards,
*Submission 2886 Authors*

---

### Meta-Review · Area_Chair_RCww · 2026-01-07

**Summary:**

The paper proposes AdaViewPlanner, a two-stage framework for automating viewpoint planning in 4D scenes by leveraging pre-trained video diffusion models. The primary concerns are regarding the method's generalization capabilities (specifically its dependence on a single backbone and performance on real-world data vs. synthetic data), the justification for specific architectural choices (e.g., the motion injection module), and the rigor of the evaluation metrics (specifically the stability of MLLM-based scoring and user study details). The reviewers generally found the approach novel and the results strong, with the rebuttal effectively addressing the concerns.

**Reviewer Concerns:**

Generalization & Backbones: The authors successfully demonstrated that their method is not backbone-specific by providing additional results using the open-source Wan2.2-5B model.

Real-World Applicability: Concerns about the method's reliance on synthetic data were resolved by adding evaluations on real-world datasets (AMASS and GTA-Human).

Design Justification: The request for ablation studies was met with new comparisons verifying the effectiveness of the "Spatial Motion Attention" mechanism over other injection methods.

Evaluation Rigor: The authors clarified the user study methodology and demonstrated the stability of their MLLM-based metrics by reporting standard deviations.

**Reviewer Scores:**

The authors have done an overall great job in addressing the concerns raised by the reviewers and imo the reviewers have reached a strong consensus to accept the paper after the rebutal so I see no reason to reject the paper.

---

### Decision · Program_Chairs · 2026-01-26

Accept (Poster)